# The *Legionella* collagen-like protein employs a distinct binding mechanism for the recognition of host glycosaminoglycans

Saima Rehman [1], Anna Katarina Antonovic[2], Ian E. McIntire[3], Huaixin Zheng[3], Leanne Cleaver [1], Maria Baczynska[1,4], Carlton O. Adams[3], Theo Portlock [1,5], Katherine Richardson[5], Rosie Shaw[5], Alain Oregioni[6], Giulia Mastroianni[2], Sara B-M. Whittaker[7], Geoff Kelly[6], Christian D. Lorenz [4], Arianna Fornili [2] ✉, Nicholas P. Cianciotto [3] ✉ & James A. Garnett [1] ✉

Bacterial adhesion is a fundamental process which enables colonisation of niche environments and is key for infection. However, in *Legionella pneumophila*, the causative agent of Legionnaires' disease, these processes are not well understood. The *Legionella* collagen-like protein (Lcl) is an extracellular peripheral membrane protein that recognises sulphated glycosaminoglycans on the surface of eukaryotic cells, but also stimulates bacterial aggregation in response to divalent cations. Here we report the crystal structure of the Lcl C-terminal domain (Lcl-CTD) and present a model for intact Lcl. Our data reveal that Lcl-CTD forms an unusual trimer arrangement with a positively charged external surface and negatively charged solvent exposed internal cavity. Through molecular dynamics simulations, we show how the glycosaminoglycan chondroitin-4-sulphate associates with the Lcl-CTD surface via distinct binding modes. Our findings show that Lcl homologs are present across both the Pseudomonadota and Fibrobacterota-Chlorobiota-Bacteroidota phyla and suggest that Lcl may represent a versatile carbohydrate-binding mechanism.

*Legionella pneumophila* is a Gram-negative bacterium that inhabits both natural and artificial freshwater systems. It thrives within a complex aquatic microbiome, which includes other biofilm associated bacterial species and cyanobacteria[1,2]. It infects and replicates within amoebae and ciliates[3] but as an opportunistic pathogen it can also infect the lungs and causes Legionnaires' disease, and the self-limiting and milder Pontiac fever[4]. Infection occurs via inhalation of water droplets from contaminated sources, where it invades macrophages in the lungs and replicates intracellularly, resulting in

pneumonia[5]. During infection *L. pneumophila* first binds the eukaryotic cell-surface, then after cell entry, it evades degradation through the formation of a specialised membrane bound replicative compartment, the *Legionella* containing vacuole (LCV)[6]. *L. pneumophila* utilises a type IV secretion system (T4SS/Dot/Icm) to transport >300 effectors directly into the host cytoplasm, which are key factors that drive LCV biogenesis and bacterial replication[7,8]. In addition, *L. pneumophila* employs a type II secretion system (T2SS/Lsp) to export >25 substrates/effectors out of the bacterium, and these play major

[1]Centre for Host-Microbiome Interactions, Faculty of Dental, Oral & Craniofacial Sciences, King's College London, London, UK. [2]Department of Chemistry, School of Physical and Chemical Sciences, Queen Mary University of London, London, UK. [3]Department of Microbiology and Immunology, Northwestern University Feinberg School of Medicine, Chicago, IL, USA. [4]Biological Physics & Soft Matter Research Group, Department of Physics, King's College London, London, UK. [5]School of Biological and Behavioural Sciences, Queen Mary University of London, London, UK. [6]The Medical Research Council Biomedical NMR Centre, The Francis Crick Institute, 1 Midland Road, London NW1 1AT, UK. [7]School of Cancer Sciences, College of Medical and Dental Sciences, University of Birmingham, Birmingham, UK. ✉e-mail: a.fornili@qmul.ac.uk; n-cianciotto@northwestern.edu; james.garnett@kcl.ac.uk

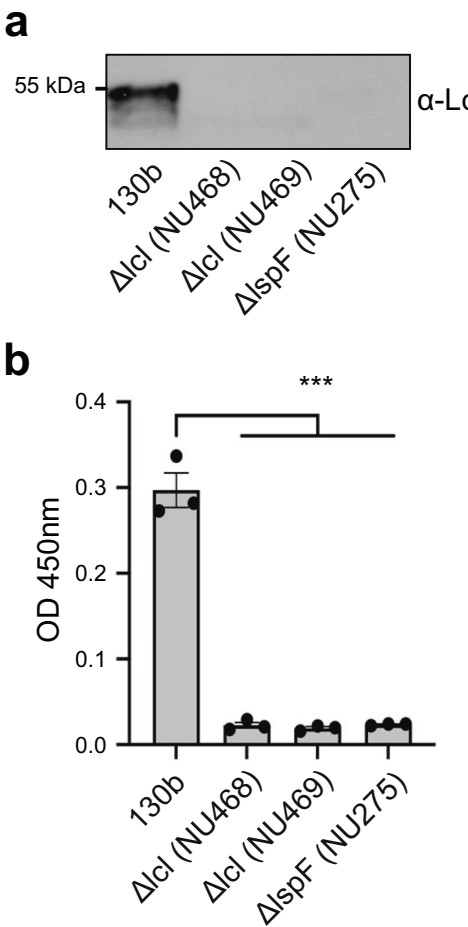

**a**

55 kDa

α-Lcl

130b
Δlcl (NU468)
Δlcl (NU469)
ΔlspF (NU275)

**b**

***

OD 450nm

130b
Δlcl (NU468)
Δlcl (NU469)
ΔlspF (NU275)

**Fig. 1 | *L. pneumophila* 130b surface association of Lcl. a** Analysis of Lcl secretion from BYE culture supernatants of wild-type 130b, *lcl* mutants NU468 and NU469, and *lspF* mutant NU275 reacted with anti-Lcl antibodies. Results are representative of two independent experiments. **b** Detection of bacterial surface binding. Whole cell ELISA of wild-type 130b, *lcl* mutants NU468 and NU469, and *lspF* mutant NU275 detected with anti-Lcl antibodies. Comparison of NU468, NU469, and NU275 to 130b shows a strong significant difference by two-tailed Student's *t* test (***$p < 0.0001$). Data are presented as mean values ± standard error of mean (SEM) derived from $n = 3$ biologically independent experiments. OD optical density. Source data are provided as an accompanying Source Data file.

roles in supporting the early stages of infection and during extracellular survival[9–18].

We initially identified the *Legionella* collagen-like protein (Lcl) in a proteomic study of type-II dependent secretion in *L. pneumophila* strain 130b[11]. Subsequently, the *lcl* gene was detected in >500 other *L. pneumophila* strains examined, indicating that Lcl expression is a conserved trait of the *L. pneumophila* species[19–21]. Although limited in its broader prevalence within the *Legionella* genus, relative to that of other T2SS substrates, the *lcl* gene occurs in five out of 57 other *Legionella* species examined (i.e., *Legionella oakridgensis*, *Legionella nagasakiensis*, *Legionella hackeliae*, *Legionella quateirensis* and *Legionella tucsonensis*) and the majority of these are associated with human infection[9]. In addition to being detected in culture supernatants on multiple occasions[11,19,22], Lcl appears to also be a peripheral membrane bound protein and upon its secretion from *L. pneumophila* it is targeted to the bacterial surface[19,23] and found in outer membrane vesicles[22]. Lcl is important for *L. pneumophila* auto-aggregation and biofilm formation[20,23–25], although this precise mechanism remains unclear. However, Lcl can also facilitate adhesion and cell entry of *L. pneumophila* to human lung epithelial (A549), lung mucoepidermoid

(NCI-H292), and macrophage (U937) cell lines, and this indicates that Lcl has a fundamental role during infection of lung tissue[19,20].

Lcl contains both an N-terminal region composed of collagen-like repeat (CLR) sequences, which are variable in length between different strains, and a C-terminal region with no overall sequence homology outside of the *Legionella* genus[11,19,26]. The C-terminal region of Lcl binds a range of sulphated glycosaminoglycan (GAG) polysaccharides that are present within the lung[20], including heparin and chondroitin-4-sulfate, while the collagen-like region has been shown to bind fucoidan[21], a heavily sulphated GAG found in many species of brown seaweed. GAGs are diverse linear carbohydrate structures that are formed from repeating disaccharide units of an amino sugar (*N*-acetylglucosamine or *N*-acetylgalactosamine) and glucuronic acid or galactose[27]. Sulphated GAGs exist as protein conjugates in the plasma membrane of nucleated cells and secreted into the extracellular matrix, and many bacterial pathogens including *L. pneumophila* use host GAGs as a means of adhesion during infection[28–30]. However, this is not well understood in *L. pneumophila* and there is a general lack in our structural and mechanistic understanding of cellular adhesion across the *Legionella* genus, which is a key step during colonisation and host invasion.

In this study, we report a structural model for full-length Lcl based on X-ray crystallographic, in silico modelling and nuclear magnetic resonance (NMR) spectroscopic data. We show that Lcl is also targeted to the surface of *L. pneumophila* 130b strain after its secretion, and this is mediated by its N-terminus. We present the crystal structure of the Lcl C-terminal domain (Lcl-CTD) which reveals an unusual trimer arrangement, and our structural and biochemical studies demonstrate a distinct GAG binding mechanism. Our work provides a molecular understanding of how Lcl can recognise and interact with a broad range of GAG ligands and provides strong evidence for the role of Lcl in facilitating direct recognition of glycosaminoglycans in host tissue during *L. pneumophila* infection.

## Results

### Lcl is expressed on the surface of *L. pneumophila* strain 130b
Previously, immunoblot analysis identified Lcl in an outer membrane fraction of wild-type strain Philadelphia-1[19], and an immuno-fluorescence assay detected the protein on the surface of strain Lp02, which is a lab-generated derivative of Philadelphia-1[23]. Since Lp02 contains multiple point mutations and a large (~45-kb) deletion in the bacterial chromosome[31], we began this study by determining whether Lcl is also surface-exposed in wild-type strain 130b in addition to being present within its culture supernatants. To that end, Lcl from strain 130b (numbered 1 to 401 from the mature protein; ORF *lpw28961*; *lpg2644* in strain Philadelphia-1, *lpp2697* in strain Paris)[9,11] was expressed recombinantly in *Escherichia coli*, purified, and then used to generate polyclonal anti-Lcl antibodies. In confirmation of our earlier work[11], immunoblot analysis revealed Lcl in the culture supernatants of strain 130b but not in the supernatants of either a T2SS (*lspF*) mutant or two constructed *lcl* mutants (Fig. 1a). By utilising a whole-cell enzyme-linked immunosorbent assay (ELISA) method that had previously examined the location of another T2SS-dependent protein, ChiA[18], we determined that Lcl is in fact present on the surface of wild-type strain 130b but not the *lspF* mutant or *lcl* mutants (Fig. 1b).

### Overall architecture of Lcl
We next turned our attention to the structural characterisation of Lcl. Using Multi-Angle Light Scattering (MALS), we determined a molecular mass of $123.4 \pm 0.2$ kDa (theoretical mass 42.3 kDa) for recombinant Lcl (Fig. 2a), which supported Lcl being a trimer in solution. Inspection of the Lcl sequence from strain 130b indicated three defined regions: a collagen-like repeat (CLR) region (consensus repeat: GPQGLPGPKGD(K/R)GEA) and C-terminal region (CTD) which contains a domain of unknown function (DUF1566)[32], but also a 30 residue N-terminal helical

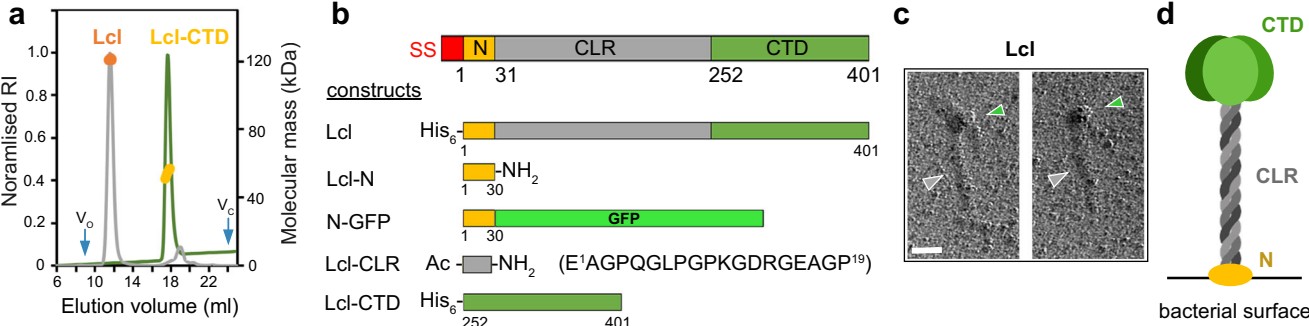

**Fig. 2 | Global characterisation of Lcl. a** Size-exclusion chromatography coupled to multi-angle light scattering (SEC-MALS) profile of recombinant Lcl and Lcl-CTD, using a Superose 6 Increase 10/300 column. Normalised refractive index (grey and green line) and average molecular weight calculated across the elution profile (orange and gold line) are shown for Lcl and Lcl-CTD, respectively. Void ($V_o$) and column ($V_c$) volumes are highlighted. RI refractive index. **b** Schematic of the Lcl domains with residue numbering based on mature sequence shown below. SS periplasmic signal sequence, N N-terminal helix, CLR collagen-like repeat region, CTD C-terminal domain, GFP green fluorescent protein, $His_6$ 6×histine tag, Ac N-terminal peptide acylation, $NH_2$ C-terminal peptide amidation. Lower: constructs used in this study with position of $His_6$ affinity tags shown. Peptide modifications are annotated (Ac acylation, $NH_2$ amidation) along with sequence and numbering. GFP green fluorescent protein. **c** Micrograph showing lollipop-shaped structures of Lcl trimers. The concentration of Lcl was 5 μg/ml. Scale bar = 50 nm. The globular shapes correspond to trimeric C-terminal domains (green arrow), while the stalks contain trimeric collagen-like region (grey arrow). **d** Schematic of the Lcl trimer presented on the bacterial surface.

region (N)[33] (Fig. 2b and Supplementary Table 1). Recombinant Lcl was analysed by rotary shadowing electron microscopy and inspection of the micrographs revealed a clear "lollipop-shaped" structure with a globular head and a stalk, consistent with a trimer of C-terminal domains and a triple helical collagen-like region, respectively (Fig. 2c, d and Supplementary Fig. 1).

### Structural features of Lcl-N

Examination of the N-terminal region of Lcl (Lcl-N) using neural network-based modelling and solution $^1H$ NMR nuclear Overhauser effect spectroscopy (NOESY), suggested that it forms an amphipathic helix[34] and can bind to sodium dodecyl sulfate (SDS) micelles (Fig. 3a, b and Supplementary Fig. 2). This indicated that it may act as an extracellular membrane anchor for Lcl after its secretion, and so we next assessed its ability to bind to the surface of *L. pneumophila*. We created an Lcl-N GFP fusion (N-GFP) (Fig. 2b) and using size exclusion chromatography (SEC)[17] observed both a major monomer species and a minor trimer species, but with the trimer population increasing with increased N-GFP concentration (Fig. 3c). Monomeric N-GFP was isolated and incubated with wild-type strain 130b and showed significant binding when compared with GFP alone (Fig. 3d). Using AlphaFold2[35] we produced consistent models of an Lcl-N trimer, where three parallel helices pack together through burial of their conserved hydrophobic face (Val10, Val14, Leu17, Leu21, Ile25) (Fig. 3e and Supplementary Fig. 2), and with other conserved residues localised to the N-terminal interface (Ser4) or contributing to a charged surface (Lys13, Lys18, Lys24). Three replica molecular dynamics (MD) simulations of 1 μs each were then run on the top ranked model to assess the stability of the trimer (Supplementary Table 2). While Lcl-N maintained the overall structure over the time course (Fig. 3f), we observed that one of the helices acts as a bridge between the other two, which share less interactions with one another, and this provides a potential mechanism for the exchange between monomer and trimer (Supplementary Fig. 2).

### Structural features of Lcl-CLR

We next probed the quaternary structure of the collagen-like region using standard multidimensional NMR. We designed a peptide that encompassed a consensus repeat sequence (Lcl-CLR) and observed a monomeric species that formed a polyproline II conformation in solution at 15 °C (Fig. 2b and Supplementary Fig. 3). When an Lcl-CLR peptide containing uniformly $^{15}N$ labelled glycine residues was studied

at 2 °C, a $^1H$-$^{15}N$ heteronuclear single quantum correlation (HSQC) spectrum showed six glycine cross-peaks from the monomeric peptide, but also a higher molecular weight species containing at least 16 distinct glycine residues (Fig. 2f). This suggested that the Lcl-CLR peptide is also in equilibrium between a monomeric and pseudo-symmetric trimer state under these conditions, containing six and 18 glycine residues, respectively. This was further supported by the comparison of cross-peaks in NOESY and rotating frame Overhauser effect spectroscopy (ROESY) spectra, where there were significant differences between NOE/ROE patterns for the higher molecular weight species at 2 °C, which disappeared at 37 °C (Supplementary Fig. 4). Analysis of intact Lcl using circular dichroism (CD) spectroscopy showed negative and positive peaks at 199 nm and 222 nm, respectively, which is in line with previous reports for Lcl from the LpO2 strain[21], and is indicative of a collagen-like structure (Supplementary Fig. 5). Furthermore, while monitoring the peak at 199 nm over increasing temperatures, we observed a two-stage unfolding process with $T_m$ values of 38 and 45 °C. Together this further supports the CLR region of Lcl forming a triple helical structure in solution.

### Overall structure of Lcl-CTD

As anticipated, analysis of the Lcl C-terminal domain (Lcl-CTD, residues 252 to 401) with MALS again revealed a stable trimer in solution (55.0 ± 0.1 kDa; theoretical mass 18.6 kDa) and crystallographic studies were initiated. The structure of Lcl-CTD was determined using selenomethionine single wavelength anomalous dispersion (Se-SAD) phasing, with electron density maps refined to 1.9 Å (Supplementary Table 3). Lcl-CTD is composed of a trimer with disordered N-termini (Glu252 to Val270) that could not be modelled, with each domain having an identical conformation formed from two α-helices (H2, H3), two $3_{10}$-helices ($3_{10}1$, $3_{10}2$) and nine β-strands (S1–S9) (Fig. 3a, b and Supplementary Fig. 6). Using small angle X-ray scattering (SAXS) we confirmed that the crystal structure is consistent with solution measurements and that the N-terminus can form several conformations, with an $R_g$ value of 2.7 nm and a $D_{max}$ of 9.7 nm (Supplementary Figs. 7, 8 and Supplementary Tables 4, 5).

The Lcl-CTD structure is stabilised through the burial of an unusually small surface area per subunit (~14,000 Å$^2$) and this is due to a solvent accessible cavity permeating from the underside into the core of the trimer (Fig. 3c). Inter-subunit interactions are mediated by charge complementary (e.g. Asp316, Asp319, Arg342) and hydrophobic residues (e.g. Trp315, Ile321, Phe343) and while the internal

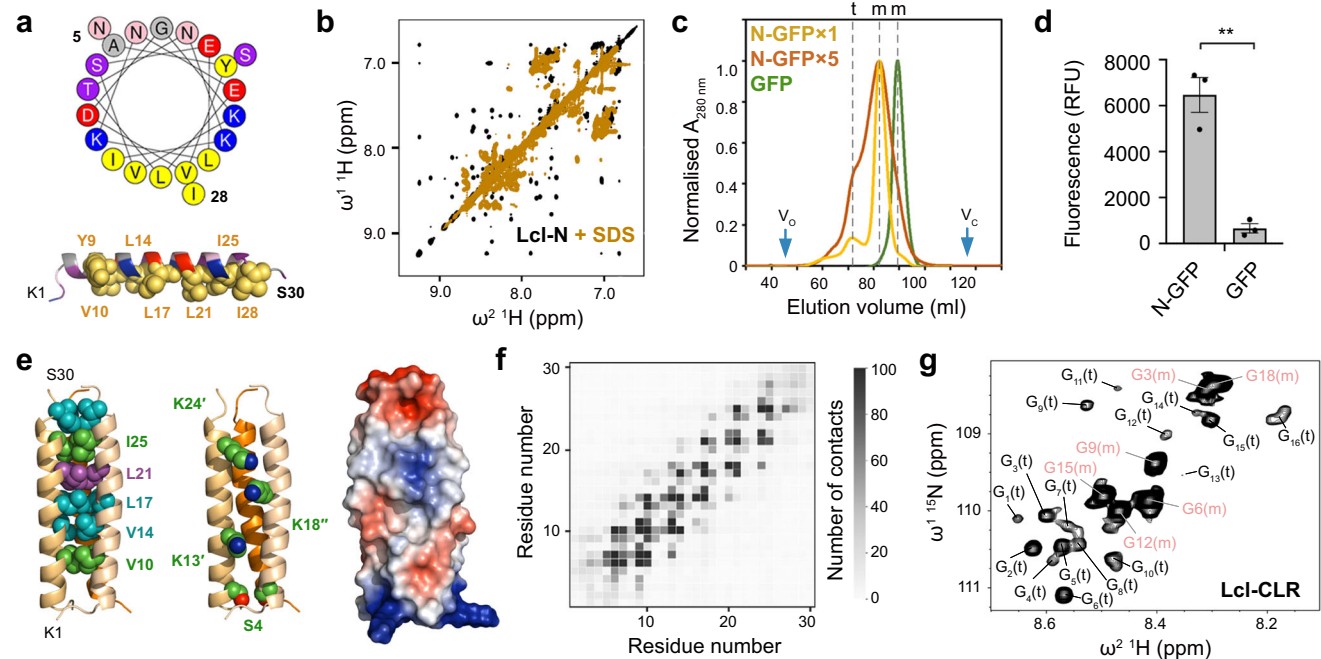

**Fig. 3 | Structural features of the Lcl-N-terminus and collagen-like repeat region. a** Structural model of Lcl-N and helical wheel diagram generated by HELIQUEST[34], with terminal residues numbered. Yellow/grey: large/small hydrophobic, pink/purple: large/small polar, blue/red: positively/negatively charged. **b** $^1H$-$^1H$ NOESY spectra of Lcl-N peptide in the presence/absence of 80 mM per-deuterated SDS, highlighting the amide region. **c** SEC profile of recombinant N-GFP and GFP, using a S200 column. Normalised absorbance (280 nm) across the elution profile is shown for N-GFP loaded at 2.5 mg/ml (N-GFP × 1; wheat), 12.5 mg/ml (N-GFP × 5; orange), and GFP (2.5 mg/ml; green). Void ($V_o$) and column ($V_c$) volumes are highlighted, as are monomeric (m) and trimeric (t) species. **d** Binding of purified N-GFP and GFP to the *L. pneumophila* 130b surface, detected through GFP fluorescence. Comparison of N-GFP to GFP shows a strong significant difference by two-

tailed Student's *t* test (**$p < 0.01$). Data are presented as mean values ± SEM derived from $n = 3$ biologically independent experiments. Source data are provided as an accompanying Source Data file. **e** AlphaFold2 model of trimeric Lcl-N, with conserved residues shown as spheres (teal: 100% identical; green: >50% identical; purple: inserted sequence). **f** Map of the mean number of contacts between residues (centre of mass) from different monomers, within a 10 Å cut-off, during a 1 µs MD simulation of the Lcl-N trimer model. Source data are available at https://doi.org/10.5281/zenodo.10961237. **g** $^1H$-$^{15}N$ HSQC spectrum of $^{15}N$-glycine labelled Lcl-CLR peptide showing resonances for monomeric (m) and trimeric (t) states. Assignment of specific glycines residues in monomeric Lcl-CLR is shown with peak positions for trimeric Lcl-CLR glycine residues numbered from left to right in subscript.

surface contains negatively charged patches (e.g. Asp336, Glu368), the upper surface displays strong positive charge (e.g. Arg342, Lys369, Lys380, Lys385 and Lys391) (Fig. 3d, e). Using the DALI server[36] we established that the Lcl-CTD monomer is similar to C-type lectin-like domains found in snake venom toxins and bacterial invasins/intimins[37–40]. However, Lcl-CTD lacks disulfide bonds and the expected motifs required for $Ca^{2+}$/carbohydrate and integrin/Tir binding (Supplementary Fig. 9), and we could not identify any trimeric structures that share tertiary homology.

The DUF1566/pfam07603[32] domain is found in diverse proteins from a wide range of prokaryotes. DUF1566 is also located between residues 314 to 399 of Lcl-CTD and is composed of the H2, $3_{10}1$, H3 and $3_{10}2$ helices, and S5-S9 strands (Supplementary Fig. 10). With truncation of the S1-S4 β-strands almost all inter-subunit interactions are still maintained, but with the depth of the internal cavity of Lcl-CTD greatly reduced. Highly conserved residues in the DUF1566 domain are largely located within the core of Lcl-CTD, with just three residues located on the surface: Trp315 at the inter-domain interface, and Arg338 and Glu344, which form an intra-domain salt bridge within the internal cavity. While a generic role for the DUF1566 domain is not clear, based on Lcl it could act in carbohydrate recognition and/or promote trimer formation. Further examination of DUF1566 containing proteins that also possess a collagen-like repeat region (gly_rich_SclB superfamily) shows Lcl actually belongs to a larger family, with homologues identified in *Legionella bononiensis* and *Legionella longbeachae* from the *Legionella* genus, but also in species across the Pseudomonadota phylum (*Comamonas sp.*, *Methylomonas paludism*, *Methylobacter sp.*, *Thiocystis minor*), and

the Fibrobacterota-Chlorobiota-Bacteroidota (FCB) superphylum (*Bacteroidetes bacterium*, *Bacteroidia bacterium*, *Candidatus Fluviicola riflensis*, *Chitinophagaceae bacterium*, *Flavobacteria bacterium*, *Formosa sp.*, *Fluviicola sp.*, *Nonlabens sp.*, *Oceanihabitans sediminis*, *Psychroflexus planctonicus*, *Winogradskyella pacifica*, and *Winogradskyella wichelsiae*) (Supplementary Data 1).

## GAGs bind the charged surface of Lcl-CTD

Chondroitin is composed of repeating disaccharide units of [−4] GlcA($\beta$1-3)GalNAc($\beta$1-]$_n$ (GlcA: D-glucuronate; GalNAc: *N*-acetyl-D-galactosamine), with chondroitin-4-sulfate (C4S) sulphated at the C4 position of GalNAc[27]. Heparin is formed of repeating disaccharide units of [−4]IdoA($\beta$1-4)GlcN($\beta$1-]$_n$ (IdoA: L-iduronate; GlcN: D-glucosamine) and is highly sulphated, with sulphation at the 2O position of IdoA (IdoA(2S)) and the 6O and N positions of GalNAc (GlcNS(6S)) being the most common form[27]. Both C4S and heparin are abundant in the lung[41] and have variable molecular weights that range between ~5–50 kDa, which equates to ~15–135 disaccharide repeats in each GAG chain. Intact Lcl was previously shown to recognise a range of variable length commercially prepared sulphated GAGs, including C4S and heparin (Fig. 4a), with the isolated C-terminal domain of Lcl also showing binding to heparin[20]. We therefore attempted to crystallise Lcl-CTD in the presence of defined C4S (GlcA/GalNAc(4S)) and heparin (IdoA(2S)/GlcNS(6S)) fragments with 4, 6 and 8 disaccharide repeats (degree of polymerisation; dp4, dp6, dp8, respectively) but were unsuccessful. Nonetheless, we did identify a crystal form of Lcl-CTD grown from high concentrations of ammonium sulfate and solved its structure using molecular replacement and refined electron density maps to 1.9 Å

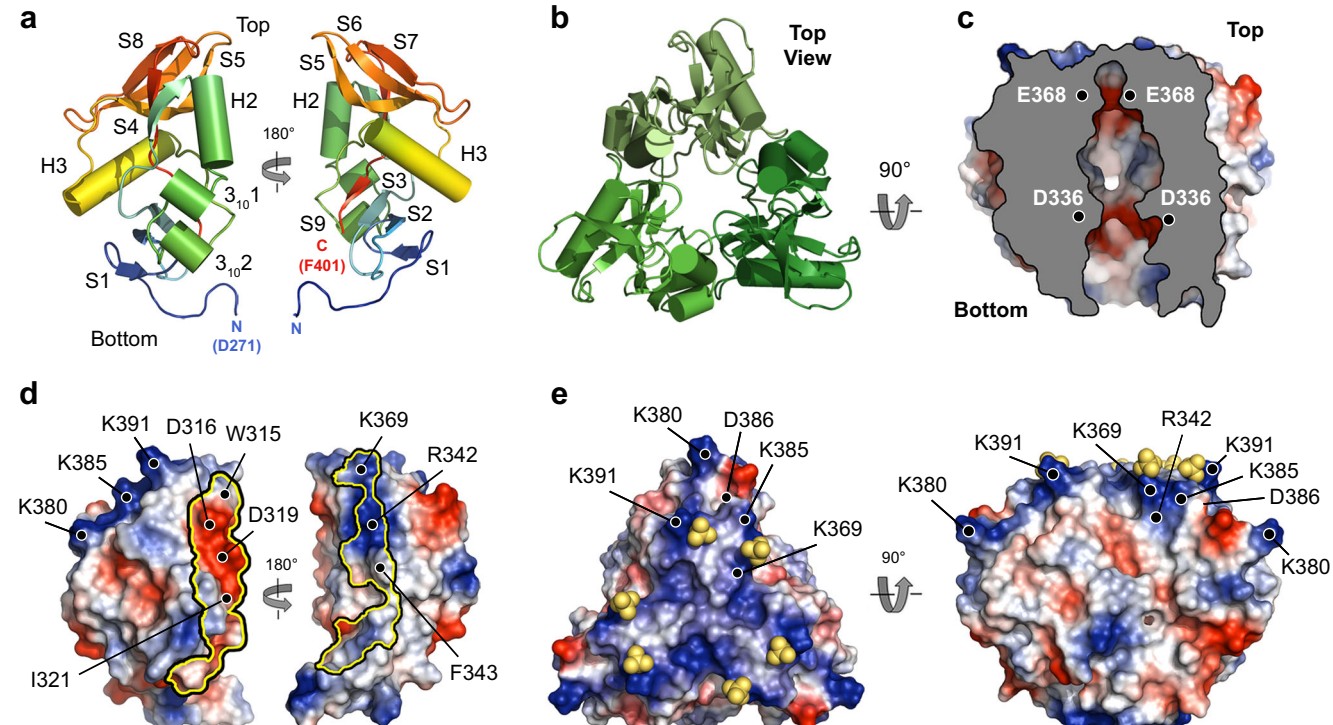

**Fig. 4 | Crystal structure of Lcl-CTD. a** Monomer of Lcl-CTD shown as cartoon and rotated by 180°. **b** Trimer of Lcl-CTD shown from the top as cartoon. **c** Trimer of Lcl-CTD shown from the side as a cut-away electrostatic surface highlighting the internal charged cavity. Position of Asp336 and Glu368 in two chains is shown. **d** Monomer of Lcl-CTD shown as electrostatic surface and rotated by 180°, with the inter-trimer interface highlighted with a yellow outline. Inter-trimer residues and charged surface residues are highlighted. **e** Crystal structure of trimeric Lcl-CTD/SO₄ shown as electrostatic surface and highlighting the bound sulfate ions (yellow spheres) and charged surface residues.

(Supplementary Table 3). The two trimer structures are highly similar (Root Mean Square Deviation (RMSD) over all $C_\alpha$ atoms of 0.3 Å) (Supplementary Fig. 11) but in this form, two sulfate ions were also observed on the surface bound to residues Lys369 and Lys391 (Fig. 3e). As GAG binding sites are usually formed from clefts or relatively flat positively charged patches[42], we speculated that Lys369 and Lys391, along with the adjacent Arg342, Lys380 and Lys385 residues, may recognise the negatively charged sulfate groups that decorate GAG polymers.

To assess this GAG binding model, we created constructs carrying R342A, K369A, K380A, K385A or K391A mutations (Lcl-CTD^R342A, Lcl-CTD^K369A, Lcl-CTD^K380A, Lcl-CTD^K385A and Lcl-CTD^K391A, respectively) which we anticipated would abrogate binding to GAGs. In addition, we also created constructs carrying a D386A mutation (Lcl-CTD^D386A) located on the Lcl-CTD surface, and a E368A mutation (Lcl-CTD^E368A) within the internal cavity, which we expected would not affect binding. Using SAXS, all constructs except for Lcl-CTD^R342A produced scattering profiles like wild-type Lcl-CTD, confirming that they were still correctly folded (Supplementary Fig. 12 and Supplementary Table 4). However, Arg342 forms intermolecular hydrogen bonds between Asp316 and Asp319 on the adjacent chain, and the R342A mutation resulted in destabilisation into monomer/trimer (3:1 ratio) (Supplementary Fig. 12), and so this construct was not used for subsequent analysis. We then assessed the ability of the correctly folded mutants to bind immobilised commercially prepared C4S and heparin extracted from bovine trachea and porcine intestinal mucosa, respectively, using an ELISA method. As anticipated, constructs carrying the K369A, K380A, K385A or K391A mutations all displayed a significant reduction in their ability to bind these GAGs when compared with wild-type Lcl-CTD, while the Asp386 mutation showed no difference. However, the Glu368 mutation resulted in higher binding capacity (Fig. 4b).

## C4S binds Lcl-CTD across multiple domains

Although the structures of C4S and dermatan sulfate differ in just the location of hydroxyl and carboxyl groups at the C2 and C5 positions of D-glucuronate and D-iduronate, respectively, Lcl does not bind dermatan sulfate[20,43]. In an attempt to understand this specificity, we started by using solution NMR spectroscopy to investigate interactions between Lcl-CTD and C4S. Using a partially deuterated sample and multidimensional transverse relaxation-optimised spectroscopy (TROSY) NMR we were able to assign 61% of the potential amide backbone resonances of Lcl-CTD (Supplementary Fig. 13). Most missing assignments were located at the N-terminus (Glu252 to Ser275), the H2 helix (Asp316 to Asn323; positioned at the inter-domain interface), and the adjacent S3-S4 loop and S4 strand (Val303 to Ser311) (Fig. 5a). Furthermore, many peaks displayed variable intensity and ~10% of residues were present in multiple conformational states (Supplementary Fig. 13). We then compared ¹H-¹⁵N TROSY spectra of Lcl-CTD titrated against increasing concentrations of commercially prepared C4S and observed significant broadening that approached saturation at 0.5 mg/ml C4S (Supplementary Fig. 14). Although no reliable data could be measured for Lys369, Lys380, Lys385 and Lys391 due to spectral overlap, significant peak broadening was observed for the neighbouring residues Thr381 and Thr392 (Fig. 5b–d). Moreover, substantial broadening was also detected for residues adjacent to Lys369 (Tyr292, Thr313, Trp315, His326, Arg342, Met350).

We next carried out molecular docking with HADDOCK[44,45], using monomeric and trimeric Lcl-CTD and dp4, dp6, dp8 and dp10 C4S oligosaccharides as starting structures, and ambiguous interaction restraints (AIRs) derived from the GAG binding ELISA and NMR chemical shift perturbations (CSP). Docking between monomeric Lcl-CTD and C4S dp8 (cluster one of the three major clusters) produced models consistent with the experimental data (Supplementary Fig. 15), and a trimer bound with one molecule of C4S dp8 was then created (HM

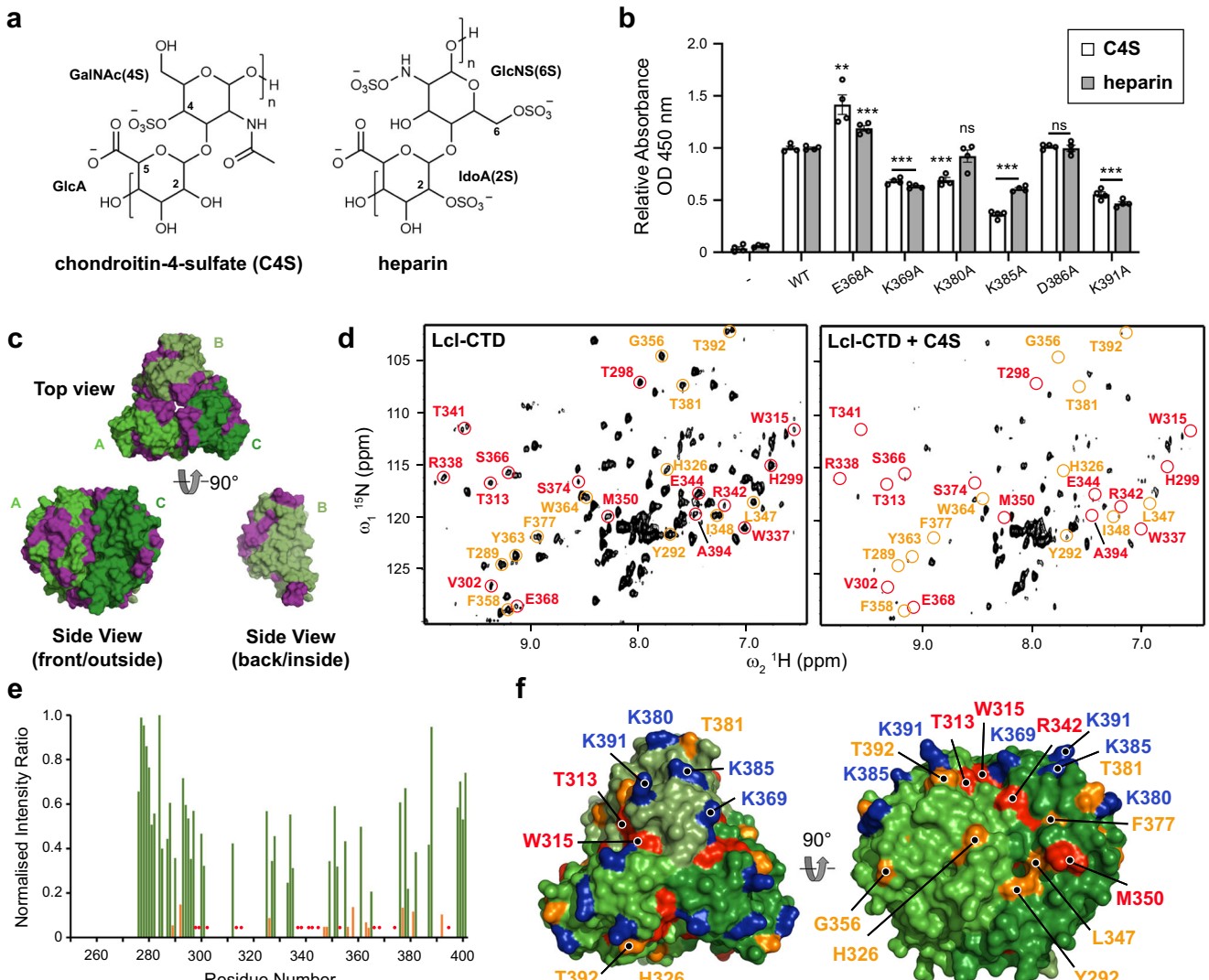

**Fig. 5 | GAG binding to Lcl-CTD. a** Chemical structure of chondroitin-4-sulfate (C4S) and heparin. GlcA: D-glucuronate; GalNAc(4S): N-acetyl-D-galactosamine-4-O-sulfate; IdoA(2S): α-L-iduronate-2-O-sulfate; GlcNS(6S): 6-O-sulpho-2-(sulphoamino)-D-glucosamine. **b** ELISA analysis of binding between immobilised mixed length C4S or heparin and wild-type (WT) and mutant (E368A, K369A, K380A, K385A, D386A, K391A) His-tagged Lcl-CTD. BSA-coated wells (−) were used as controls. Comparison of mutated Lcl-CTD to their respective WT shows a strong significant difference by two-tailed Student's $t$ test (***$p < 0.001$: K368A (heparin), K369A (C4S/heparin), K380A (C4S), K385A (C4S/heparin), K391A (C4S/heparin); **$p < 0.001$: K368A (C4S)) except for K380A (heparin; $p = 0.241$) and D386A (C4S/heparin; $p = 0.588/0.931$). Data are presented as mean values ± SEM derived from $n = 4$ biologically independent experiments. OD optical density. Source data are

provided as an accompanying Source Data file. **c** Trimer of Lcl-CTD shown as surface representation with residues whose amides could be assigned coloured green, and those that could not be assigned coloured purple. **d** NMR $^1$H–$^{15}$N TROSY spectrum of Lcl-CTD in presence (right) or absence (left) of 0.5 mg/ml mixed length C4S. Chemical shifts that have disappeared after addition of C4S are highlighted in red, and those that display significant broadening (reduction of >85% peak intensity) are highlighted in orange. **e** Same information as (**d**) shown as a bar graph with orange bars highlighting significant peak broadening on addition of C4S. Missing assignments have a value of zero and those where peaks disappear on addition of C4S are highlighted with red circles. Source data are provided as an accompanying Source Data file. **f** As (**d**) and (**e**) but mapped onto the surface trimer of Lcl-CTD.

model) and further examined using MD (Fig. 6a and Supplementary Fig. 16). Docking between trimeric Lcl-CTD (HT1 and HT2 models) caused changes to the trimer interface and they were unstable during MD simulations (Supplementary Fig. 16), and not taken forward for further analysis. MD simulations were also run on Lcl-CTD (residue 271 to 401) alone (Supplementary Fig. 17). Analysis of the Root Mean Square Fluctuation (RMSF) profiles indicated a high flexibility for the first 7 residues of each monomer (Asp217 to Ile223), consistent with the disordered nature of the adjoining N-terminal part of the chain. However, the overall trimeric structure of the domain was stable throughout the simulations, with RMSD from the initial structure quickly reaching a plateau and staying below 2 Å in all the replicas.

While the overall structure of Lcl-CTD was found to be stable in all replicas of simulations run on the HM model, C4S displayed highly dynamic binding to the Lcl-CTD surface (Supplementary Fig. 16). Although during the simulations C4S remained in contact with Lcl-CTD for much of the time, the different components of the glycan frequently detached and then reattached to different regions of the Lcl-CTD surface. From its starting position, during the simulations the polysaccharide either remained in the same region or explored other parts of the top surface of the protein. As indicated by the spatial distribution function (sdf) of C4S sulfur atoms (Fig. 6b) and the frequency of contacts between C4S and the Lcl-CTD trimer, (Fig. 6c), C4S more often bound to the central region of the top surface.

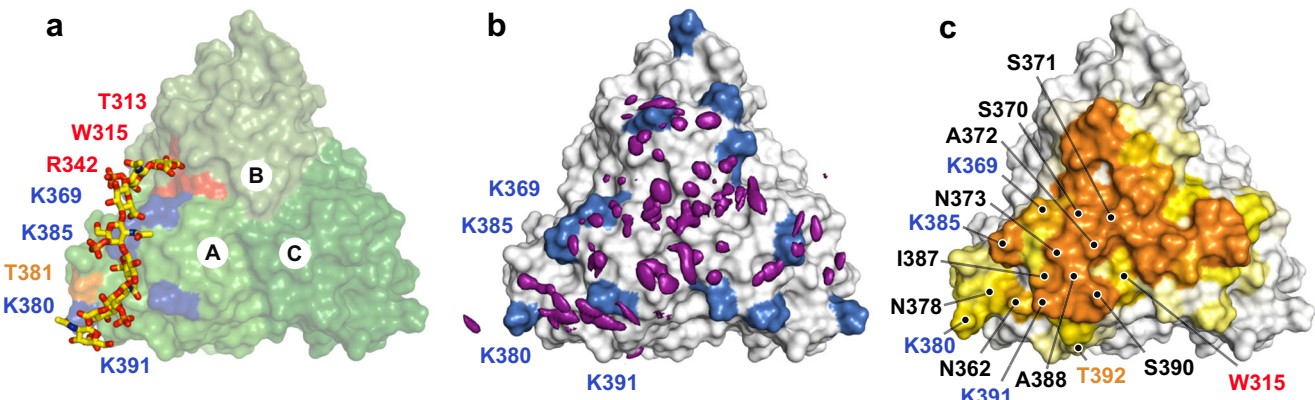

**Fig. 6 | Molecular dynamics analyses of C4S dp8 binding to Lcl-CTD. a** Modified HADDOCK model (HM: C4S dp8 docked against a monomer of Lcl-CTD and reconstituted as a timer) used as a starting structure for MD simulations. Surface residues in close contact of C4S are annotated and coloured red (reduction >100% peak intensity by NMR), orange (reduction >85% peak intensity) or blue (lysine residues identified by ELISA). Chains are labelled A to C. **b** Spatial distribution function (sdf) of the sulfur atoms of C4S dp8 during the simulations. The purple isosurface connects the points with sdf = 20 × average value. The protein surface (initial MD structure) is represented in white with the positions of Lys369, Lys380, Lys385 and Lys391 coloured blue. **c** Frequency of occurrence (occupancy) of contacts between C4S dp8 and the Lcl-CTD during the simulations colour-mapped onto the protein surface (initial MD structure) from white (0–10%) to orange (40.9%). Residues with an occupancy >10% in chain A are annotated as (**a**) or black (identified from MD). Source data are available at https://doi.org/10.5281/zenodo.10974841.

Structures from all the replicas were clustered using an optimised cut-off of 17.5 Å on the pairwise $C_\alpha$ RMSD values, the high value reflecting the variety of binding poses explored by C4S in the different replicas. Three major C4S binding modes were identified, and although they had a relatively high RMSD, they broadly reflected a preference for Lcl-CTD surface localisation of the bound glycan chain, although the clusters did not reflect a preference in orientation (Fig. 7a). After considering the 3-fold symmetry of the system, the first and third mode were found to be closely related, with C4S dp8 showing a similar position in the two modes, and these were therefore combined. The first and major binding mode is the most frequently observed (M-BM; 63% frequency) and represents C4S binding to the top, central region of Lcl-CTD, along the chain A/C interface. The second and minor binding mode (m-BM; 20% frequency) represents C4S binding primarily to Lcl-CTD chain A and resembles the initial input HM model (Fig. 6a).

A more detailed analysis of the distance and interactions between C4S and Lcl-CTD highlighted that C4S can bind across one (36% frequency), two (35% frequency) or all three (28% frequency) Lcl-CTD chains (Fig. 7b, Supplementary Fig. 18 and Supplementary Tables 6, 7). While binding of C4S to a single chain of Lcl-CTD is observed in both the major and minor binding modes, binding across multiple Lcl-CTD chains largely reflects the major binding mode alone. Scrutiny of these different complex formations indicates that the Lcl-CTD residues found to be most frequently involved in hydrogen bonding with C4S are Ser371, Ser390 and Lys391, which are located at the central region of the top surface (Fig. 6c and Supplementary Table 7). On the other hand, we observed that C4S dp8 binds to Lcl-CTD using 4 to 6 saccharide units (GalNAc(4S) and GlcA), either as a continuous stretch or with the glycan looped out in the middle of the chain, and forms hydrogen bonding interactions through its carboxylates, amides, sulfates, and hydroxyl groups. Moreover, examination of the representative structures of C4S dp8 binding across 1-, 2-, and 3-chains of Lcl-CTD suggests that the replacement of D-glucuronate with D-iduronate would result in the disruption of some hydrogen bond interactions (Fig. 7b). It would also require changes in the glycan conformation to avoid clashes within dermatan sulfate and between dermatan sulfate and Lcl-CTD, and together this provides at least some explanation for the selectivity of Lcl-CTD for different GAGs.

## Discussion

Adhesion is a fundamental process in bacteria and adhesin proteins often work in synergy to enable colonisation of niche environments. Several adhesins have been identified in *L. pneumophila* that play important roles in the recognition of eukaryotic hosts, and these include the type IV pilus (T4P)[46] and its associated PilY1 pilus tip adhesin[47,48], Hsp60[49], RtxA[50], MOMP[51], LaiA[52], and Lcl[19]. We have determined that Lcl is a trimeric structure formed of three regions: an N-terminal helix/coiled-coil, an elongated collagen-like region, and a DUF1566 containing C-terminal region. We previously observed Lcl secreted in bacterial culture supernatants of *L. pneumophila* strain 130b[11]. Subsequently Lcl was detected on the bacterial surface in strains Philadelphia-1[19] and Lp02[23], and we have now shown this in 130b. The *L. pneumophila* T2SS exports >25 proteins, and three of these associate with host organelles and/or the bacterial surface upon their secretion (i.e., ChiA, Lcl, ProA); we previously observed ChiA and ProA tethered to the LCV membrane[53] and ChiA to the outer membrane surface[18]. Although the mode of ProA membrane binding is unclear, ChiA binds the *L. pneumophila* surface using its N3 domain, formed of a fibronectin III module domain-like fold[18]. In addition, we have shown that NttA binds phosphatidylinositol-3,5-bisphosphate $(PtdIns(3,5)P_2)$ and other phosphorylated phosphoinositides, which indicates that NttA may also be targeted to host organelles during infection[17]. In this study we have revealed that Lcl binds the bacterial surface through its N-terminal region. Using a GFP fusion, we observed that Lcl-N is predominantly a monomeric amphipathic helix, but it can also form a coil-coiled structure with increasing concentration. Together with our MD analysis this suggests that within intact Lcl, Lcl-N will primarily form a coil-coiled, but this is not symmetrical, and as binding was shown using the isolated monomeric species, the trimer may become displaced and bind as three independent amphipathic helices. Alternatively, purified Lcl-N will inevitably still contain trimers and a trimeric binding mechanism to the bacterial membrane or to another yet to be identified outer-membrane structure cannot be ruled out; however, the surface of the current trimer model is not hydrophobic and would not be able to bind within a lipid-membrane in its current conformation. Nonetheless, this represents a distinct mechanism that has not been observed for other T2SS substrates.

Bacterial collagen-like proteins have been identified in a wide range of Gram-positive and Gram-negative bacteria[54]. A defining

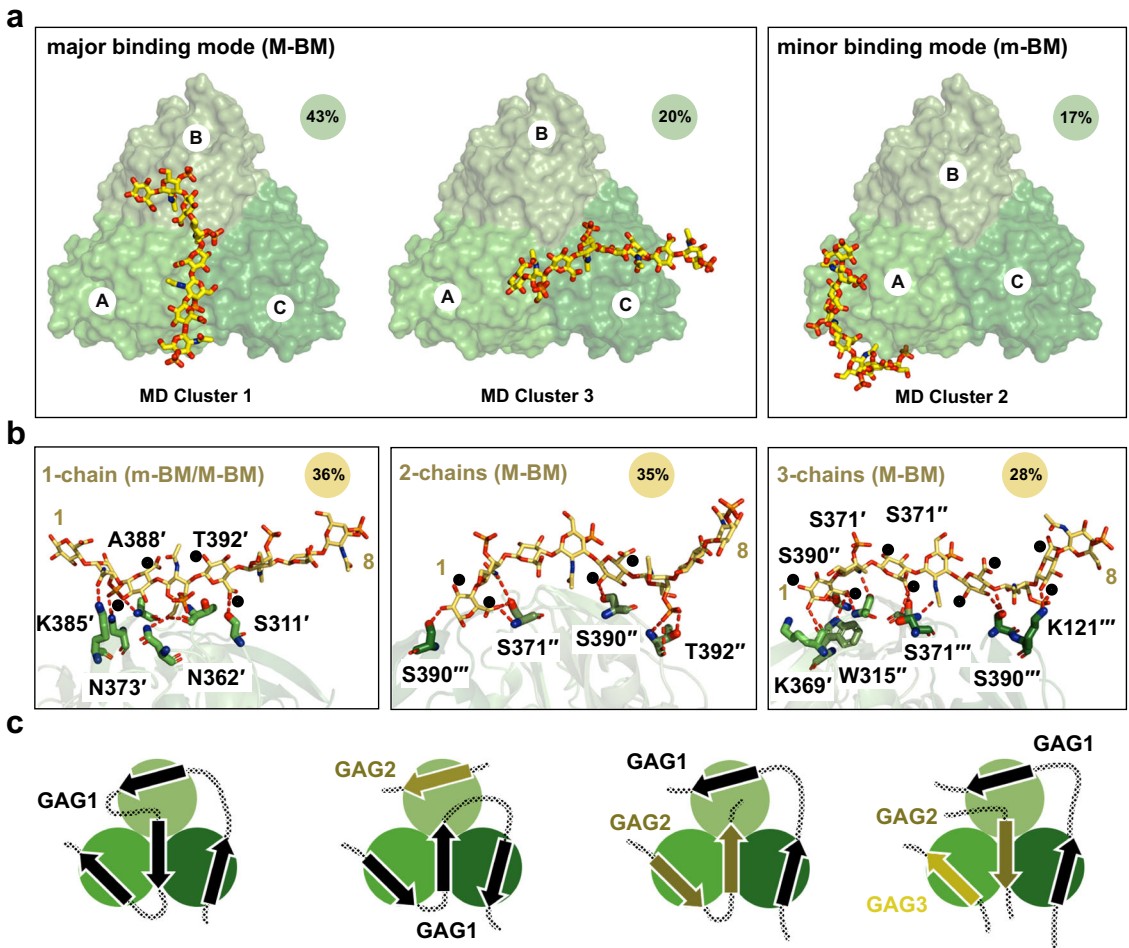

**Fig. 7 | Binding modes of C4S dp8 to Lcl-CTD. a** Representative C4S dp8 structures (cluster centre) of the first (MD cluster 1, population = 43%), second (MD cluster 2, population = 21%), and third (MD cluster 3, population = 20%) most populated clusters are shown as sticks, together with the initial protein structure (green surface). The position and orientation of cluster 3 is like that of cluster 1 when 3-fold rotational symmetry is considered. These therefore represent two major binding modes: clusters 1 and 3 (M-BM, major binding mode, population = 63%) and cluster 2 (m-BM, minor binding mode, population = 21%). Chains are labelled A to C. **b** Representative MD structures of C4S dp8 bound to Lcl-CTD selected to illustrate binding across 1-chain (population = 36%; derived from m-BM and M-BN), 2-chains (population = 35%; primarily derived from M-BM) and 3-chains (population = 28%; primarily derived from M-BM). Structures were selected from

replica 7, 11 and 6, respectively. Hydrogen bonding interactions between C4S and Lcl-CTD detected by PLIP[110] are shown as dashed red lines. The protein residues involved in the interactions are labelled. Cyan spheres indicate the location of C1 hydroxyl and C5 carboxyl groups within C4S D-glucuronate residues, which if switched would perturb binding. **c** Models of glycosaminoglycan (GAG) binding to the Lcl-CTD trimer. Schematics of the general major and minor binding mode of C4S are shown with bound glycan chain as an arrow, which can bind in either direction. The Lcl-CTD surface could support simultaneous binding to GAGs from one continuous chain (black connected arrows) and/or from multiple chains (olive and wheat arrows). Source data are available at https://doi.org/10.5281/zenodo.10974841.

feature of collagen is the presence of Gly-X-Y repeats, where in eukaryotes X and Y are often proline and hydroxyproline, respectively, with hydroxyproline mediating inter-chain hydrogen bonding to sta-bilise the triple helical structure. However, bacteria cannot make hydroxyproline, and their collagen-like structures do not have a requirement for proline. Instead, they contain a higher proportion of charged/polar residues, and these are predicted to interact across different chains[55,56]. As both bacterial and eukaryotic collagens display similar thermal stabilities ($T_m$ ~ 35–39 °C and ~37 °C, respectively)[55], it is unclear why eukaryotic systems do not produce bacterial-like collagen, although these structures may be unfavourable for the formation of higher-order fibrils, which are not observed in prokaryotes.

From our examination, Lcl from *L. pneumophila* strain 130b con-tains 12 repeats of a consensus 15 residue sequence (GPQGLPGPKGD(K/R)GEA) within its collagen-like region. Analysis of Lcl from other strains isolated from clinical samples, the environment, and hot springs, however, has demonstrated a high polymorphism within this region, with Lcl from Philadelphia-1 containing 19 repeats[19]. While hot spring

isolates (≥40 °C) displayed a preference for 13 repeats, clinical and environmental isolates (≤37 °C) were bimodal with a preference for both 8 and 13/14 repeats. Using a 19-residue peptide encompassing a single Lcl CLR consensus sequence, we observed this peptide to be largely monomeric, but able to form triple helical structures with a reduction in temperature. Together this suggests that variability in Lcl repeats may reflect the minimal length of collagen required for Lcl to retain its folding under different environmental temperatures. Our CD spectroscopy analysis showed that Lcl has two melting temperatures of 38 °C and 45 °C (Supplementary Fig. 5), which are similar to other reported bacterial and eukaryotic collagens[55]. However, in eukaryotes, folding of collagens are initiated at their C-terminus and mediated by trimerisation domains, before being propagated through to the N-terminus[57]. This indicates that the Lcl C-terminal domain may also function to initiate folding of the collagen-like region.

Lcl has been shown to mediate adhesion/invasion of *L. pneumo-phila* to a range of host cell types. In one study, a *lcl* mutant displayed a ~30% reduction in binding to NCI-H292 lung mucoepidermoid cells,

compared with the wild-type Lp02[20]. In another study, incubation of wild-type Philadelphia-1 with Lcl antibodies resulted in a ~50% drop in binding to A549 lung epithelial cells, and 0–30% drop in binding to U937 macrophage cells, although no difference in binding was observed with the amoeba *Acanthamoeba castellanii*[19]. Specifically, Lcl binds to sulphated GAGs on the surface of host cells and both the collagen-like and C-terminal regions have been implicated here[19–21]. When *lcl* containing 14 or 19 repeats was expressed in Philadelphia-1, there was an increase in adhesion/invasion of A549 cells with 14 compared with 19 repeats, but the opposite was observed with U937 cells[19]. Fucoidan has also been shown to bind the collagen-like repeat region of Lcl from Lp02 with a higher affinity than to the C-terminal domain, and increasing the number of CLRs has been correlated with tighter binding[21]. Together this suggests that the C-terminal domain plays a general role in the recruitment of ligands, but at least for some GAGs, synergistic binding along the collagen-like region can provide further increases in overall affinity.

The C-terminal domain of Lcl is highly conserved (>97%) across different strains of *L. pneumophila* and we have determined that it forms a distinct trimer structure with a deep negatively charged internal cavity and positively charged external surface (Fig. 3). Intact Lcl from Lp02 binds fucoidan with 10-fold higher affinity than C4S ($K_D$ 18 nM and 173 nM, respectively)[43], and this likely reflects the increased level of sulphation in fucoidan. Using mixed chain length heparin and C4S, we have demonstrated that the strong positive charge on the Lcl-CTD surface is important for glycan recognition (Fig. 4b). Furthermore, using MD simulations, and focussing on binding to a dp8 structure of C4S, we have identified two predominant binding modes for this ligand (Fig. 7a). A major mode (M-BM) which runs across the middle of the Lcl-CTD trimer, and a minor mode (m-BM) which is largely localised to a single chain of the trimer. Only Lys385 and Lys391 form hydrogen bonds with C4S during the simulations, with relatively low frequency, while Lys369, Lys380, Lys385 and Lys391 are all within proximity (Fig. 6b and Supplementary Tables 6, 7). This indicates that the primary role of these lysine residues is to provide general electrostatic attraction for the GAGs, rather than specific recognition. Furthermore, during the simulations the majority of the Lcl-CTD upper surface is involved in sampling dp8 C4S, mainly through serine, threonine, and asparagine residues (Fig. 6c), with C4S repeatably dissociating and reassociating at different sites. This indicates that C4S does not bind at two distinct sites but is instead recognised through a range of conformational states, albeit preferentially within two regions. Such fuzzy binding has been observed with intrinsically disordered proteins, for example Cdc4 binding to multiply phosphorylated Sic1[58], and this may reflect a mechanism that enables recognition of a broad range of ligands. Residues at the inter-trimer interface (Ser371, Ala372, Lys391) appear to have a more targeted role in binding, although it is not clear whether these are specific for C4S or represent a more general GAG binding mode. As C4S and other GAGs can contain up to ~135 disaccharide repeats it is feasible that one or more glycan chains could bind simultaneously at multiple sites on the Lcl-CTD surface (Fig. 7c).

Using NMR with mixed chain length heparin and C4S, we also observed major CSPs for residues on the side of the Lcl-CTD trimer (i.e., Arg342, Met350) (Fig. 5d), although this binding was not observed during the MD simulations using C4S dp8. Arg342 and Met350 could represent a lower affinity site that is only occupied once GAGs have bound to the top surface of Lcl-CTD but may facilitate single GAG chain binding between the C-terminal domain and the collagen-like repeat region. However, we also showed that Arg342 hydrogen bonds with Asp316 and Asp319 in an adjacent chain and stabilises the trimer, and an R342A mutation produced a mixed population of monomeric (major species) and trimeric (minor species) Lcl-CTD (Supplementary Fig. 12). Furthermore, in $^1$H-$^{15}$N TROSY spectra of Lcl-CTD, many interfacial residues were either broadened out and could not be assigned (e.g. Asp316, Asp319) or were present as multiple peaks (e.g. Arg338, Thr341, Glu368) (Fig. 5a and Supplementary Fig. 13). This demonstrates that Lcl-CTD experiences conformational exchange and may be present in more than one trimeric state, although as this was not observed during the 400 ns MD simulations it must occur on the slow NMR time scale (μs to ms). Furthermore, the R342A SAXS scattering profile deviates from the wild-type at $q > 0.15$ Å$^{-1}$ (Supplementary Fig. 12 and Supplementary Table 4) indicating an alternate confirmation of the Lcl-CTD trimer, but with the same overall global structure. Therefore, as Arg342 and Met350 are located at the domain interface, CSPs observed for these residues during NMR titrations with heparin and C4S may instead reflect indirect binding events due to stabilisation of one trimeric state upon association with GAGs.

We have now identified the *lcl* gene in eight *Legionella* species, however, except for *L. quateirensis* we see little conservation in the surface lysine residues that are present in *L. pneumophila* (Supplementary Fig. 19). Although this suggests that Lcl-CTD from these other species will not exhibit a large positively charged surface, we do see conservation of other key C4S binding residues mainly located at the inter-trimer interface (i.e. Ser370, Ser371, Ala372, Asn373, Asn378). This may indicate that the Lcl C-terminal domain has different glycan specificity outside of *L. pneumophila*. Lcl is also known to mediate auto-aggregation and biofilm formation of *L. pneumophila* in the presence of divalent cations[20,25] and it has been suggested that trimeric Lcl from Lp02 can form higher-order structures which could function in clumping adjacent bacteria[21]. However, these observations were independent of divalent cations, and our biophysical characterisation of Lcl from 130b shows that it is extremely stable and homogenous (Fig. 2a, c and Supplementary Fig. 5). Glu368 is highly conserved across the *Legionella* genus and is located within the internal cavity, where three residues are in proximity perpendicular to the trimer 3-fold axis (Fig. 3c and Supplementary Fig. 19). When Glu368 was mutated to alanine, we observed a significant increase in binding of Lcl-CTD to both heparin and C4S (Fig. 3c), which again could be explained by this mutation stabilising one of the trimeric states and priming it for GAG recognition. We speculate that Glu368 may bind divalent cations and have a role in modulating the biofilm activity of Lcl, potentially through increasing the population of the alternate Lcl-CTD conformation, but further studies are needed.

## Methods

### Bacterial strains and media

All strains used in this study are listed in Supplementary Data 2. *L. pneumophila* strain 130b (American Type Culture Collection [ATCC] strain BAA-74; also known as strain AA100 or Wadsworth) served as wild type and parent for all mutants[59]. The *L. pneumophila lspF* mutant NU275 strain[60] and all newly constructed mutants (NU468, NU469) were routinely grown at 37 °C on buffered charcoal yeast extract (BCYE) agar or in buffered yeast extract (BYE) broth[61]. Isotopically defined M9 minimal medium (pH 7.4) contained (per litre) 6.0 g Na$_2$HPO$_4$·7H$_2$O, 3 g KH$_2$PO$_4$, 0.5 g NaCl, 0.12 g MgSO$_4$·7H$_2$O, 22 μg CaCl$_2$, 40 μg thiamine, 8.3 mg FeCl$_3$·6H$_2$O, 0.5 mg ZnCl$_2$, 0.1 mg CuCl$_2$, 0.1 mg CoCl$_2$·6H$_2$O, 0.1 mg H$_3$BO$_3$ and 13.5 mg MnCl$_2$·6H$_2$O, supplemented with 2 g [U-$^{13}$C$_6$]glucose and/or 0.7 g $^{15}$NH$_4$Cl (Cambridge Isotope Laboratories). M9 media was made up in deuterium oxide (Sigma) to produce perdeuterated protein samples and pH was adjusted using 1 M NaOH solution.

### Mutant construction

All primers and plasmids used in this study are listed in Supplementary Data 3 and 4, respectively. To make the *L. pneumophila* NU468 and NU469 mutant strains that have a nonpolar, unmarked deletion within the *lcl* gene, we employed overlap extension PCR (OE-PCR) followed by allelic exchange, as before[13,62]. DNA fragments of the 5′ and 3′ regions flanking the *lcl* ORF were PCR-amplified from 130b DNA using the

primer pairs *lcl*-UpF and *lcl*-UpR for 5′ *lcl*, and *lcl*-DownF and *lcl*-DownR for 3′ *lcl*. A kanamycin (Kn)-resistance gene flanked by Flp recombination target sites was PCR-amplified from the vector pKD4[63] using the primers *lcl*-P1 and *lcl*-P2. We then performed two-step OE-PCR to combine the 5′ and 3′ regions of *lcl* with the respective Kn-resistance cassette. A PCR product matching the correct target size was gel purified and ligated into pGEM-T Easy (Promega) to yield plcl::Kn. After transforming strain 130b with the newly made plasmid, bacteria containing an inactivated *lcl* gene was obtained by plating on BCYE agar containing Kn. Confirmation of the mutated *lcl* gene was done by PCR using the above-mentioned primers. Following transformation with pBSFLP, which encodes a Flp recombinase along with a gentamicin-resistance marker[63], mutants harbouring the desired unmarked deletion and lacking pBSFLP were recovered by plating on BCYE agar containing 5% (w/v) sucrose and scored for loss of resistance to both Kn and gentamicin, as before[64]. The mutants were verified by sequencing of PCR amplicons.

## Immunoblot analysis of bacterial culture supernatants
Wild-type and mutant *L. pneumophila* strains that had been grown for 3 days on BCYE agar were suspended into BYE broth to an $OD_{660}$ of 0.3 and grown overnight at 37 °C to an $OD_{660}$ of -1.5. Supernatants were obtained by centrifugation, sterilised by passage through 0.2-μm filters (EMD Millipore), and then concentrated, as before[53]. Following dilution in SDS–loading buffer, the samples were subjected to PAGE and immunoblot analysis. To that end, purified recombinant Lcl protein (above) was submitted to Lampire Biological Laboratories (Pipersville, PA) at a concentration of 2 mg/ml for the production of rabbit polyclonal antisera, analogously to what we had been done before for other secreted proteins of *L. pneumophila*[53]. Following an overnight incubation at 4 °C in 5% BSA (w/v)–Tris-buffered saline (TBST), the blot was incubated overnight at 4 °C with the primary anti-Lcl antiserum at 1:1000 in 5% BSA-TBST. After four, 10-min washes with the TBST buffer, the membrane was further incubated for 1 h at room temperature with secondary goat anti-rabbit horseradish peroxidase antibody (Cell Signaling Technology, Catalog #704) at 1:10,000 in 5% BSA-TBST. Finally, after another series of washes, the blot was developed using Amersham ECL Prime reagent and exposed to X-ray film, as before[13,18]. The full scan blot is provided as an accompanying Source Data file.

## Bacterial whole-cell ELISA
The assay for detecting protein on the surface of *L. pneumophila* was done as previously described for the detection of ChiA, another substrate of the *L. pneumophila* T2SS[18]. The bacterial strains (130b, NU468, NU469, NU275) were grown on BCYE agar for 3 days at 37° and then using a sterile cotton swab, bacteria were resuspended in 1 ml sterile PBS to an $OD_{660}$ 0.3. These were centrifuged at $10,000 \times g$ for 3 min, and then washed once with PBS. Bacteria were fixed in 4% (w/v) paraformaldehyde for 10 min, followed by two 1 ml washes in PBS, and then resuspension in coating buffer (100 mM bicarbonate/carbonate buffer, pH 9.6) to a final $OD_{660}$ 0.03. In total, 100 μl of this suspension were added into the wells of Nunc MaxiSorp immunoassay plates (Thermo Fisher Scientific) and incubated overnight at 4 °C. The wells were then washed three times with 200 μl of wash buffer (PBS, 0.05% Tween-20), followed by addition of 200 μl of blocking buffer (wash buffer with 5% dried milk) and incubation for 1 h at 25 °C. Blocking buffer was removed and then samples were incubated with 100 μl of primary anti-Lcl antibody (Lampire Biological Laboratories, Pipersville, PA) at 1:1000 dilution in blocking buffer for 1 h at 25 °C. Following three, 200-μl washes with wash buffer, samples were incubated with 100 μl of secondary goat anti-rabbit horseradish peroxidase antibody (Cell Signaling Technology, Catalog #704) diluted 1:1000 in blocking buffer for 1 h at 25 °C. Following five washes with 200 μl wash buffer, samples were incubated with 100 μl 3,3′,5,5′-Tetramethylbenzidine

(TMB) substrate for 15 min at 25 °C, and then, the reaction was stopped by addition of 50 μl of 2 N sulfuric acid. Absorbance values were measured at 450 nm with wavelength correction of 570 nm using a microplate reader (Synergy H1, BioTek).

## Construction of recombinant expression plasmids
Intact *lcl* (residues 1–401) and its C-terminal fragment (residues 252–401) were amplified by PCR from *L. pneumophila* 130b gDNA using primer pairs RLC1/RLC2 and RLC3/RLC4, respectively. These were then cloned into the pET-46 Ek/LIC vector (Novagen) using ligation-independent cloning. Synthetic genes gRLCm1 to gRLCm7 (Synbio Technologies, https://synbio-tech.com), pNGFP and pGFP (GenScript, https://www.genscript.com), were cloned into pET28b vector using NcoI and XhoI restriction sites to create plasmids pRLCm1 to pRLCm7, pNGFP and pGFP, respectively. All plasmids and synthesised genes used in this study are listed in Supplementary Data 4 and 5, respectively.

## Protein purification
Intact Lcl, N-GFP and GFP were expressed in *E. coli* strain BL21(DE3) (New England Biolabs) grown in LB media containing either 50 μg/ml ampicillin (Lcl) or 50 μg/ml kanamycin (N-GFP and GFP). Lcl-CTD was expressed in *E. coli* strain BL21(DE3) (New England Biolabs) grown with 50 μg/ml ampicillin in either LB media, minimal media supplemented with selenomethionine (Molecular Dimensions), minimal media containing 0.07% (w/v) $^{15}NH_4Cl_2$ (Cambridge Isotope Laboratories), 100% (v/v) $D_2O$ (Sigma) or minimal media containing 0.07% (w/v) $^{15}NH_4Cl_2$, 0.2% (w/v) $[^{13}C]$glucose (Cambridge Isotope Laboratories), 100% (v/v) $D_2O$. Expression was induced with 0.5 mM isopropyl-d-1-thiogalactopyranoside (IPTG) at an $OD_{600nm}$ of 0.6 and cells were harvested after growth overnight at 18 °C. Samples were purified using nickel-affinity chromatography followed by gel filtration using a Superdex-200 gel-filtration column (GE Healthcare), equilibrated in 20 mM Tris-HCl pH 8.0, 200 mM NaCl. To ensure efficient back exchange of amide protons, perdeuterated Lcl-CTD samples were initially purified in the presence of 8 M urea and then after nickel-affinity chromatography they were refolded by dialysis against 20 mM Tris–HCl pH 8, 200 mM NaCl, 1 M urea, 5 mM ethylenediaminetetraacetic acid (EDTA) and then 20 mM Tris–HCl pH 8, 200 mM NaCl. Engineered Lcl-CTD carrying R342A, E368A, K369A, K380A, K385A, D386A and K391A mutations in the *lcl-CTD* gene (Lcl-CTD^H326A, Lcl-CTD^R342A, Lcl-CTD^R368A, Lcl-CTD^K369A, Lcl-CTD^K380A, Lcl-CTD^K385A, Lcl-CTD^D386A and Lcl-CTD^K391A, respectively) were purified as wild-type Lcl-CTD.

## SEC-MALS
Lcl or Lcl-CTD were injected onto a Superose 6 Increase 10/300 column (GE Healthcare) coupled to a Wyatt Technology system and run in 20 mM 4-(2-hydroxyethyl)-1-piperazineethanesulfonic acid (HEPES) pH 7.5, 200 mM NaCl. BSA was run as a monodisperse reference protein. A dn/dc value of 0.185 ml/g was used for molecular weight calculations and data analysis was performed with Astra V software.

## Rotary shadowing electron microscopy
The overall structure of Lcl was analysed using transmission electron microscopy after rotary shadowing using an adapted mica sandwich technique[65,66]. Five μl of Lcl in 20 mM HEPES pH 7.5 (5 μg/ml) was sprayed on a freshly cleaved mica sheet, allowed to adsorb, and then washed with ultrapure water. The mica was mounted on the stage of a Polaron Freeze fracture instrument and then freeze dried at −100 °C. The temperature was lowered to −150 °C for shadowing with Pt/C on a low angle (5°) and a carbon backing layer was added for support. These were removed from the mica in distilled water and placed on 400 mesh copper grids. Micrographs were taken with a JEM 1230 transmission electron microscope operated at 80 kV.

## Peptide modelling

Modelling of monomeric and trimeric Lcl-N was carried out using the sequence for Lcl residues 1–30 from *L. pneumophila* 130b strain with AlphaFold2 or AlphaFold2-multimer[35], respectively. Sequence alignments and templates were generated through MMseqs2[67] and HHsearch[68], and run through the ColabFold notebook[69]. No prior template information was provided, and sequences used during modelling were both paired from the same species and unpaired from multiple sequence alignment.

## Peptide NMR

All peptides were synthesised by Thermo Scientific to >95% purity. Unlabelled Lcl-N peptide (KSNPASQAYVDGKVSELKNELTNKINSIPS-NH$_2$) was resuspended to 1 mM in 25 mM NaPO$_4$ pH 6.5, 100 mM NaCl, 10% (v/v) D$_2$O with or without 80 mM perdeuterated d$_{25}$-SDS, and $^1$H-$^1$H NOESY spectra (200 ms mixing time) were recorded at 298 K on a 700 MHz Bruker Avance III HD equipped with cryoprobe. Unlabelled Lcl-CLR peptide (Ac-EAGPQGLPGPKGDRGEAGP-NH$_2$) and Lcl-CLR peptide containing uniformly $^15$N labelled glycine residues (Ac-EAG̲PQG̲LP̲G̲PKG̲DRG̲EAG̲P-NH$_2$; labelled positions underlined) were resuspended to 3 mM in 20 mM HEPES pH 6.0, 50 mM NaCl, 10% (v/v) D$_2$O. Peptides were then incubated at 90 °C for 15 min and then 4 °C for 1 week. Full backbone and side chain assignments for the monomeric unlabelled peptide was achieved using standard double-resonance peptide assignment experiments ($^1$H-$^{15}$N HSQC, $^1$H-$^{13}$C HSQC, $^1$H-$^{13}$C total correlation spectroscopy (TOCSY), $^1$H-$^1$H TOCSY, $^1$H-$^1$H correlation spectroscopy (COSY), $^1$H-$^1$H ROESY with 200 ms mixing time) recorded at 288 K on a 700 MHz Bruker Avance III HD equipped with cryoprobe. In addition, $^1$H-$^1$H ROESY (200 ms mixing time), $^1$H-$^1$H NOESY (240 ms mixing time) and $^1$H-$^{15}$N HSQC spectra were recorded at 275 K, and a $^1$H-$^1$H NOESY spectrum (240 ms mixing time) was recorded at 310 K, on an 800 MHz Bruker Avance III HD equipped with cryoprobe. All spectra were processed using NMRPipe[70] and analysed using ANALYSIS[71]. Secondary structure propensity of the monomeric Lcl-CLR peptide at 288 K were calculated using the δ2D server, providing C$_\alpha$, C$_\beta$, H$_\alpha$, N, NH backbone chemical shifts[72]. All data was acquired using TOPSPIN 3.5.6.

## Molecular dynamics of the Lcl-N trimer

The top ranking AlphaFold2 model of trimeric Lcl-N was run through the CHARMM-GUI[73] solution builder server and placed in 0.15 M NaCl solution with 6.62 × 6.62 × 6.62 nm$^3$ box dimension, resulting in a total of 55 salt, 27969 water, and 1377 protein atoms. MD simulations were run using GROMACS 2019[74] and the CHARMM36 force field[75]. The system was thermalized, equilibrated, and simulated at 300 K and 1 bar pressure following the simulation protocol suggested by CHARMM-GUI[73]. From the final structure after pressure equilibration, three independent production trajectories of 1 μs, that resulted in statistically consistent configurational ensembles, were generated. Trajectories were analysed using the python package MDAnalysis[76]. The system setup for these simulations is summarised in Supplementary Table 2.

## Bacterial surface binding assay

*L. pneumophila* strain 130b was grown on BCYE agar plates (Oxoid, UK) at 37 °C aerobically for 3 days. Colonies were emulsified in 5 ml sterile PBS (Oxoid, UK) with a sterile cotton swab to OD$_{600\,nm}$ 0.3, centrifuged at 5000 × g for 10 min, and then washed once in sterile PBS to remove cell debris and unbound protein. N-GFP and GFP at 2.5 mg/ml (~80 μM) were purified by SEC and the monomeric species isolated. A 1.5 ml aliquot of resuspended cells was then incubated with either 20 μM N-GFP, 20 μM GFP, or sterile PBS for 1.5 hrs at room temperature with gentle end-over-end mixing. Following incubation, cells were pelleted by centrifugation at 12,000 × g and washed four times in sterile PBS, before being resuspended in 300 μl sterile PBS. Three 100 μl aliquots of resuspended cells were added to a 96-well microtitre plate and fluorescence intensity was measured at excitation/emission 489/508 nm (CLARIOstar Plus Microplate Reader), with fluorescence intensity being normalised against PBS only.

## Circular dichroism

Far-UV CD spectra were measured in a Chirascan (Applied Photophysics) spectropolarimeter thermostated at 10 °C. Spectra for Lcl (0.05 mg/ml) in 10 mM HEPES pH 8.0 was recorded from 260 to 195 nm, at 0.5 nm intervals, 1-nm bandwidth, and a scan speed of 10 nm/min. Three accumulations were averaged for each spectrum. For thermal denaturation experiments, Lcl (0.05 mg/ml) in 10 mM HEPES pH 8.0 was recorded at 199 nm between 10 °C and 75 °C in 1 °C increments. Each increment was recorded in triplicate and then averaged.

## Crystal structure determination

Selenomethionine labelled Lcl-CTD (Lcl-CTD; 15 mg/ml) and native Lcl-CTD (Lcl-CTD/SO$_4$; 20 mg/ml) in 20 mM Tris-HCl pH 8.0, 200 mM NaCl, 20 mM EDTA were crystallised using the sitting-drop vapour-diffusion method grown at 20 °C in either 2.0 M (NH$_4$)$_2$SO$_4$, 0.1 M Bis-Tris pH 6.5 or 20% (v/v) glycerol, 20% (w/v) polyethylene glycol (PEG) 4000, 30 mM NaNO$_3$, 30 mM Na$_2$HPO$_4$, 30 mM (NH$_4$)$_2$SO$_4$, 100 mM Bicine, 100 mM Tris pH 8.5, respectively. Crystals were briefly soaked in well solution complemented with additional 30% or 10% (v/v) glycerol, respectively, before flash freezing in liquid nitrogen. Diffraction data were collected at 100 K at beamline I03 of the Diamond Light Source (DLS), United Kingdom, with wavelengths 0.97969 Å (Lcl-CTD) and 0.97625 Å (Lcl-CTD/SO$_4$). Data were processed using XDS and scaled with AIMLESS, within the XIA2 pipeline[77–79]. For Lcl-CTD, two selenomethionine sites were located in each Lcl-CTD molecule using SHELXD[80] and then phases were calculated using autoSHARP[81] (figure of merit: acentric/centric 0.251/0.010; phasing power 0.865). After automated model building with ARPWARP[82], the remaining structure was manually built within Coot[83]. Refinement was carried out with REFMAC[84] using non-crystallographic symmetry (NCS) and translation-libration-screw (TLS) groups, and 5% of the reflections were omitted for cross-validation. For Lcl-CTD/SO$_4$, molecular replacement was carried out in PHASER[85] using a single chain of Lcl-CTD as the search model. Refinement was again carried out with REFMAC[84] using non-crystallographic symmetry (NCS) and translation-libration-screw (TLS) groups, and 5% of the reflections were omitted for cross-validation. Both structures were run through PDBREDO[86] as a final step of refinement. The quality of the Lcl-CTD and Lcl-CTD/SO$_4$ models were assessed by MolProbity[87]. Ramachandran statistics showed 97.7% and 98.8% of residues in the most favoured region and 100% and 100% in the allowed regions, respectively. Processing and refinement statistics of the final model can be found in Supplementary Table 3.

## SEC-SAXS

Data were collected at beamline B21 at the Diamond Light Source (DLS), UK[88]. 60 μl of WT Lcl-CTD, Lcl-CTD$^{R342A}$, Lcl-CTD$^{E368A}$, Lcl-CTD$^{K369A}$, Lcl-CTD$^{K380A}$, Lcl-CTD$^{K385A}$, Lcl-CTD$^{D386A}$ and Lcl-CTD$^{K391A}$ (5 mg/ml) in 20 mM Tris–HCl pH 8, 200 mM NaCl, 5 mM EDTA were applied to a Shodex KW403-4F column at 0.16 ml/min and SAXS data were measured over a momentum transfer range of $0.004 < q < 0.44$ Å$^{-1}$. Peak integration and buffer subtraction were performed in CHROMIXS[89]. The radius of gyration ($R_g$) and scattering at zero angle ($I(0)$) were calculated from the analysis of the Guinier region by AUTORG[90]. The distance distribution function ($P(r)$) was subsequently obtained using GNOM[90], yielding the maximum particle dimension ($D_{max}$). The disordered N-terminus of the Lcl-CTD crystal structure was built using MODELLER[91] and refinement of the N-terminus in the intact model against the corresponding SAXS curve was carried out with EOM2[92] with fixing of the ordered domains

in the trimer. Bead modelling was carried out using DAMMIF/DAMMIN[90], and compared with atomic structures using SUPCOMB[90]. CRYSOL[90] was used to compare models against solution SAXS curves. Processing and refinement statistics can be found in Supplementary Tables 4 and 5.

## GAG binding ELISA

Immulon 2-HB 96-well plates (VWR) were coated overnight at 4 °C with either 50 μl of heparin from porcine intestinal mucosa (Sigma) or chondroitin-4-sulfate from bovine trachea (Sigma) at 100 μg/ml in 50 mM carbonate/bicarbonate pH 9.6. Wells were blocked for 1 h at 25 °C with 200 μl of 0.1% (w/v) bovine serum albumin (BSA) in PBS−0.05% (v/v) Tween 20 and then washed once with 200 μl of incubation buffer (0.05% (w/v) BSA in PBS−0.05% (v/v) Tween 20). Wells were then incubated for 3 h at 25 °C with 50 μl of WT Lcl-CTD, Lcl-CTD$^{E368A}$, Lcl-CTD$^{K369A}$, Lcl-CTD$^{K380A}$, Lcl-CTD$^{K385A}$, Lcl-CTD$^{D386A}$ or Lcl-CTD$^{K391A}$ at 10 μM in incubation buffer. This was followed by four washes with 200 μl of incubation buffer and incubation with 50 μl of anti-His-HRP antibody (1 mg/ml; ThermoFisher Scientific, Catalog # MA1-21315-HRP), diluted 1:2000 in incubation buffer for 1 h at room temperature. After four washes with 200 μl of incubation buffer, 150 μl of o-Phenylenediamine dihydrochloride (Sigma) was added for 30 min and then data was recorded at 450 nm.

## TROSY NMR

Measurements were performed at 37 °C on a $^2$H$^{15}$N$^{13}$C-labelled Lcl-CTD sample (0.5 mM) in 20 mM HEPES pH 7.0, 50 mM NaCl, 5 mM EDTA, 10% D$_2$O on a cryoprobe-equipped Bruker Avance III HD spectrometer with 900 MHz Oxford Instruments magnet. Backbone assignments for 61% of Lcl-CTD (not including the N-terminal His-tag and proline residues) was achieved using standard double- and triple-assignment methods. NMR titration experiments were carried out on a Bruker Avance III HD 800 MHz spectrometer equipped cryoprobe. $^2$H$^{15}$N-labelled Lcl-CTD (0.2 mM) in 20 mM HEPES pH 7.0, 50 mM NaCl, 10% D$_2$O with the addition of 0, 10, 20, 50, 100 and 500 μg/ml chondroitin-4-sulfate from bovine trachea (Sigma) was used to measure $^1$H$^{15}$N TROSY spectra at 37 °C. All spectra were processed using NMRPipe[70] and analysed using the programme NMRVIEW[93]. Residues that displayed spectral overlap were not analysed for changes in peak intensity between different spectra. All data was acquired using TOPSPIN 3.2.

## Experimental driven docking

Molecular docking of C4S oligosaccharides to Lcl-CTD monomer and trimer was carried out with HADDOCK[44,45,94] modifying an approach previously used to dock heparin oligosaccharides[94]. Oligosaccharides dp4, dp6, dp8 and dp10 were generated by the GAG Builder server[95]. Active and passive residues were chosen based on the CSPs and ELISA-based mutational analysis. Topology and parameter files for the C4S oligosaccharides were generated using the PRODRG server[96]. Docking of dp2, dp4, dp8 and dp14 were performed for a 1:1 and 3:1 Lcl-CTD:C4S complex for the Lcl-CTD monomer and trimer, respectively. During initial rigid body docking a total of 1000 structures were generated, and then semi-flexible simulated annealing (SA) was performed on the best 200 structures followed by explicit solvent refinement. The final structures were clustered using a RMSD cut-off value of 7.5 Å and the clusters were sorted using RMSD and the HADDOCK score.

## Molecular dynamics of C4S binding

MD simulations were carried out starting from the crystallographic structure of Lcl-CTD and from three HADDOCK derived models of the complex between C4S and Lcl-CTD. Two models (HT1 and HT2) were obtained from HADDOCK where one molecule of C4S dp8 was docked against a trimer of Lcl-CTD (3:1 Lcl-CTD:C4S). The final model (HM) was obtained from HADDOCK with one molecule of C4S dp8 docked

against a monomer of Lcl-CTD (1:1 Lcl-CTD:C4S) but then reconstituted as a trimer (3:1 Lcl-CTD:C4S) based on the crystal structure. Simulations were performed using GROMACS 2020[74], with the Amber99SB*-ILDN[97] force field for the Lcl-CTD and GLYCAM-06j for C4S[98]. GLYCAM is one of the most commonly used force fields to simulate glycans and it is fully compatible with Amber force fields[99]. A truncated octahedral box of TIP3P[100] water molecules was used to solvate the systems, setting a minimum distance of 12 Å between the protein and the edges of the box. Residues with ionisable groups were set to their standard protonation states at pH 7. Counterions (Na$^+$ and Cl$^−$) were added to neutralise the system and reach an ionic strength of 100 mM, leading to a total of ~50,100 atoms (~14,600 water molecules). Periodic boundary conditions were applied. The equations of motion were integrated using the leap-frog method with a 2-fs time step. The LINCS[101] algorithm was used to constrain all covalent bonds in the protein, while SETTLE[102] was used for water molecules. Electrostatic interactions were evaluated with the Particle Mesh Ewald (PME) method[103] using a 9-Å distance cut-off for the direct space sums, a 1.2 Å FFT grid spacing and a 4-order interpolation polynomial for the reciprocal space sums. A 9 Å cut off was set for van der Waals interactions and long-range corrections to the dispersion energy were included.

Each system was minimised through 3 stages with 5000 (positional restraints on heavy atoms) + 5000 steps of steepest descent, followed by 2000 steps of conjugate gradient. Positional restraints on heavy atoms were initially set to 4.8 kcal/mol/Å$^2$ and they were gradually decreased to 0, while the temperature was increased from 200 to 300 K at constant volume. The system was then allowed to move freely and was subjected to equilibration in NVT conditions at T = 300 K. This was followed by equilibration under NPT conditions with T = 300 K and p = 1 bar. For these equilibration steps, the Berendsen[104] algorithm was used for both temperature and pressure regulation with coupling constants of 0.2 and 1 ps, respectively. At last, NPT equilibration was run after switching to the v-rescale thermostat[105] with a coupling constant of 0.1 ps and the Parrinello-Rahman barostat[106] with a coupling constant of 2 ps. Longer equilibrations were run for Lcl-CTD in the presence of C4S (45 ns in total for all the steps) compared to Lcl-CTD alone (6.5 ns), to allow for relaxation of the GAG binding pose. Lcl-CTD simulations were run in three replicas (400 ns for each production, for a total production time of 1.2 μs). For the Lcl-CTD/C4S system, preliminary 50 ns production runs were first carried out. The trimeric structure of Lcl-CTD was very stable with the HM model, while unbinding of monomers from the rest of the protein was observed for MT1 and HT2, so only the former model was retained for subsequent simulations. A total of 21 replicas were run for HM (150 ns production for each replica). The glycan remained in contact with Lcl-CTD in all but one replica, which was not considered for subsequent analysis, so that the overall production simulation time for Lcl-CTD/C4S was 3 μs.

Contacts between C4S and the protein were analysed with the bio3D[107] R-package. A residue was considered in contact with C4S if the minimum distance calculated over all pairs of non-hydrogen side chain atoms was lower than 4 Å. Frames sampled every 100 ps were analysed. The frequency of occurrence of a given contact was calculated as the percentage of frames in which that contact was observed. The highest occurring contact between any part of C4S and a given protein residue was calculated and averaged over all 20 replicas to give the final value. Cluster analyses were performed using the gromos[108] method implemented in GROMACS on the pseudo-trajectories generated by concatenating all the replicas for a given system (production only; 3 × 400 ns for Lcl-CTD and 20 × 150 ns for Lcl-CTD/C4S), with frames sampled every 1000 ps. For both calculations, the Lcl-CTD C$_α$ atoms, not including the flexible residues 271 to 277, were first fitted to the coordinates of the initial minimised structure. The distance between structures was calculated as the RMSD of the C$_α$ atoms (271 to 277

excluded) for the Lcl-CTD simulations and the RMSD of all C4S atoms for the Lcl-CTD/C4S simulations. Cut-off values were determined to optimise the clustering for each system and were set to 1.1 Å for Lcl-CTD and 17.5 Å for Lcl-CTD/C4S. The large value for C4S reflects the variety of binding poses explored by the GAG in the different replicas. The structure with the highest number of neighbours for each cluster (central structure) was selected as cluster representative. The population of each cluster was adjusted to consider the 3-fold symmetry of the system. For a given cluster, each frame in the cluster was first rotated by ~ 120° in both directions. Rotation was carried out by superimposing symmetrically equivalent monomers. If after rotation the C4S structure in the frame was found to be closer to a cluster representative different from the original cluster (as measured by the C4S RMSD), the frame was reassigned to that cluster. The spatial distribution function[109] (sdf) of C4S sulfur atoms around the protein was calculated by running the GROMACS *gmx spatial* tool on the pseudo-trajectory of concatenated replicas (production only), with frames sampled every 1 ps. Each frame was first fitted to the minimised starting structure using best-fit superposition of $C_\alpha$ atoms (271 to 277 excluded). A grid spacing of 0.5 Å was used for the sdf calculation. The average of non-null sdf values was calculated and the isosurface connecting points with sdf = 20 × average was analysed. Each frame of the Lcl-CTD/C4S simulations was classified into one of three binding categories (1-chain, 2-chains, or 3-chains) by calculating the number of Lcl-CTD chains within 3 Å of C4S (with the distance calculated as minimum distance between all possible pairs of non-hydrogen atoms from C4S and Lcl-CTD). The system setup for these simulations is summarised in Supplementary Table 2.

### Reporting summary

Further information on research design is available in the Nature Portfolio Reporting Summary linked to this article.

## Data availability

Atomic coordinates and structure factors files generated in this study have been deposited in the Protein Data Bank database under accession codes 8Q4E (Lcl-CTD) and 8QK8 (Lcl-CTD/SO₄). NMR assignments have been deposited in the Biological Magnetic Resonance Data Bank database under accession codes 52394 (Lcl-CTD) and 52395 (Lcl-CLR peptide). SAXS curves have been deposited in the Small Angle Scattering Data Bank database under accession codes SASDUG7 (Lcl-CTD WT), SASDUH7 (Lcl-CTD R477A trimer), SASDUJ7 (Lcl-CTD R477A monomer), SASDUK7 (Lcl-CTD E503A), SASDUL7 (Lcl-CTD K504A), SASDUM7 (Lcl-CTD K515A), SASDUN7 (Lcl-CTD K520A), SASDUP7 (Lcl-CTD D521A), SASDUQ7 (Lcl-CTD K526A). Initial and final structures from MD simulations are available at https://doi.org/10.5281/zenodo.10961237 and https://doi.org/10.5281/zenodo.10974841. The authors will provide raw data, additional information, and materials, including plasmids for protein expression, upon request. These should be addressed to J.G. Source data are provided with this paper.

## Code availability

GROMACS tools and bespoke python scripts using the MDAnalysis library were used to analyse the molecular dynamics trajectories for Lcl-N. These are freely available at https://doi.org/10.5281/zenodo.10961237. The codes used to analyse the molecular dynamics trajectories of Lcl-CTD/C4S are available upon request, which should be addressed to A.F.

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

## Acknowledgements

This work was supported by the MRC (MR/M009920/1, MR/R017662/1, MR/W000814/1) and EPSRC (1806169) to J.G., and NIH (AIO43987, AI175460) to N.C. Work was also supported by the Wellcome Trust (099185/Z/12/Z), and we thank HWB-NMR staff at the University of Birmingham for providing open access to their Wellcome Trust-funded 900 MHz spectrometer. In addition, this work was supported by the Francis Crick Institute through provision of access to the MRC Biomedical NMR Centre. The Francis Crick Institute receives its core funding from Cancer Research UK (FC001029), the MRC (CC1078), and the Wellcome Trust (CC1078). We also thank the Centre for Biomolecular Spectroscopy at King's College London for additional NMR access, funded by the Wellcome Trust (202767/Z/16/Z) and British Heart Foundation (IG/16/2/32273). This work made use of time on HPC granted via the UK High-End Computing Consortium for Biomolecular Simulation (HECBioSim), supported by EPSRC (EP/R029407/1, EP/X035603/1). We thank the beamline scientists at I03 and BL21 of the Diamond Light Source, United Kingdom. We would also like to thank Prof. Krishna Rajarathnam and Dr. K. Mohan Sepuru (UTMB) for their guidance in creating C4S input files for use in HADDOCK.

## Author contributions

Conceived and designed the experiments: S.R., A.K.A., I.M., H.Z., L.C., M.B., C.A., A.O., G.M., S.W., G.K., C.D., A.F., N.C. and J.G. Performed the experiments: S.R., A.K.A., I.M., H.Z., L.C., M.B., C.A., T.P., K.R., R.S., A.O., G.M., S.W., G.K., A.F. and J.G. Analyzed the data: S.R., A.K.A., I.M., H.Z., L.C., M.B., C.A., T.P., K.R., R.S., C.D., A.F., N.C. and J.G. Contributed reagents/materials/analysis tools: A.K.A., C.D., A.F., N.C. and J.G. Wrote the paper: S.R., A.K.A., L.C., M.B., C.A., C.D., A.F., N.C. and J.G.

## Competing interests

The authors declare no competing interests.
