## [Peer Review File · Nature Communications]

The *Legionella* collagen-like protein employs a distinct binding mechanism for the recognition of host glycosaminoglycansREVIEWER COMMENTS

Reviewer #1 (Remarks to the Author):

This paper presents a study of the structure and GAG binding of Lcl, using a wide range of experimental and computational methods. The work is carried out with good attention to detail and is described clearly, with an excellent level of experimental confirmation. The results are clear although not fully conclusive: however, they advance our understanding significantly and are thus well worth publishing. I have no major criticisms, but a few small comments:

1. The mode of binding to sulphated GAGs contains a mixture of geometries (eg Fig 7). It would be useful if the authors could comment whether they feel that there is a unique binding mode which they have been unable to determine, or whether it is genuinely fuzzy, for example as described by Forman-Kay's group for Cdc4 binding to multiply phosphorylated Sic1 (<https://doi.org/10.1073/pnas.0809222105>).
2. Line 67: should substates be substrates?
3. Line 209 presumably 0.3 Angstrom?
4. Line 314 delete "is"
5. Line 423 change were to was
6. Line 529 which beamline?
7. Line 1017 change peptides to peptide
8. Line 1025 change perdeutotared to perdeuterated
9. Line 1027 start should read of specific glycine residues
10. Line 1028 change glycines to glycine
11. Supplementary Figure 7 legend delete the comma after Kratky
12. Supplementary Figure 11 legend change is to in

Reviewer #2 (Remarks to the Author):

Report on Rehman et al. 454632_0

Rehman et al. have been studying an adhesin protein (Legionella collagen-like protein) produced by Legionella pneumophila and report here the crystal structure for its C-terminal domain (CTD). This globular domain makes up ~40% of the 401-residue collagen-like protein monomer, so called because the section from residues 31 to 252 form a glycine- and proline-rich triple helix. Consistent with this the CTD is also a trimer with a positively charged exterior and a negatively charged interior cavity. Using Molecular Dynamics simulations, the authors have illustrated several ways that this trimer can bind to sulfated glycosaminoglycans on eukaryotic cells, which helps explain the Legionella bacterium's infectivity. A bioinformatic analysis suggests that this type of adhesin appears in other bacterial phyla.

The authors have used a variety of physical and computational analyses to probe the structure of

the Legionella collagen-like protein. Electron microscopy images of this protein in Fig 2. C and SI provide strong support for the model presented in Fig. 2b by showing a globular structure (CTD trimer) on top of a linear stalk (collagen-like region). The crystal structure of the CTD trimer is detailed and shows an uneven charge distribution that is consistent with the binding of sulfated glycosaminoglycans as suggested by modeling studies.

One of the speculations about the Lcl model is its attachment to the outer-membrane of the host bacterium through an amphipathic helical region (residues 1- 31) illustrated in Fig. 2 panel d. The hydrophobic patch on one quadrant of the helix is flanked by positively and negatively charged residues, which is often seen in helix coiled coils. Have the authors scanned the helix sequence for its propensity to form a coiled coil structure with a complementary helix on the bacterial cell? Has the 31-residue alpha-helical peptide been shown to bind *L. pneumophila*?

The Discussion section contains a lot of speculation and it would be preferable to make a clearer distinction between what has been established by experimentation and what is speculated to happen. For example, regarding lines 314 and 315: “In this study we have revealed that Lcl is likely binds the bacterial surface via an amphipathic helix motif at its N-terminus, which represents a new mechanism that has not been observed for other T2SS.” Going from ‘likely binds’ to: ‘a new mechanism that has not been observed for other T2SS substrates (Fig. 2d,e)’, is a big stretch. Why not tag the amphipathic helix motif with GFP and see how well it binds to the bacteria, The trimeric structure of both the CTD and the collagen-like region (CLR) are consistent. Are there any indications of one trimerization facilitating the other? Which is likely to be the driver of oligomerization?

Figure 2. In panel a, the void and total volumes of the size-exclusion column should be indicated in the chromatogram. In panel b, the schematic of the Lcl trimer could show CLR region forming a triple helix rather than parallel strands.

Given the similarity of the Lcl-CTD monomer to C-type lectin-like domains, could you comment on the number and location of disulfide bonds in the fold? C-type lectin-like domains typically have two or three highly conserved disulfide bonds.

Minor points

Line 314: needs revision

Line 317: Gram-positive, Gram-negative

Paragraph beginning line 413: replace apostrophes with prime symbols.

Some hyphens missing: 1-nm bandwidth; 9-Å distance; carbohydrate-binding mechanism; ligation-independent

Reviewer #3 (Remarks to the Author):

Authors present experimentally validated model of Legionella collagen-like protein (Lcl) in interaction with sulphated glycosaminoglycans(GAG). This interaction is responsible for the colonization of Legionella pneumophila, but also stimulates bacterial aggregation in response to divalent cations.

By applying the tour de force of multiple experimental technique (MALS, SAXS, Rotary shadowing electron microscopy, MX, NMR-NOESY, CD) and in situ modeling (AlphaFold2 prediction of Lcl- N-term, GAG-Lcl docking, MD simulation) authors provide accurate model of Lcl- C-terminal domain /GAG interactions. Most importantly reported the single mutation study of Lcl-CTD further support the model Lcl-CTD – GAG interactions.

Well planned experiment provided a good data quality that support main conclusion of the article. The article is well written and it was a pleasure to read the sequential story that start with identifying whether the Lcl is expressed on the surface of *L. pneumophila* and finished with model of how GAG binds Lcl-CTD across multiple domains.

In more details authors report the crystal structure of the Lcl C-terminal domain (Lcl-CTD) where SAXS and NMR show that Lcl-CTD forms an unusual dynamic trimer arrangement with a positively charged external surface and a negatively charged solvent exposed internal cavity. Through MD and docking, they show how the GAG associates with the Lcl-CTD surface via unique binding modes. I do not have any major concern only a few recommendations that follow below:

Page 9, 219: Please comment on the role of R342A in the dissociation of the trimer assembly. The author should discuss the specific impact of the R342A mutation on the trimer assembly. Does it lead to the destabilization or disruption of the trimer structure? Any experimental findings or insights on this mutation's effect should be elaborated upon.

Page 10 and Abstract: "Lcl-CTD trimer experiences conformational exchange and is a dynamic structure" - please elaborate on the nature and type of dynamicity.

In the text, there's a statement about the dynamic nature of the Lcl-CTD trimer, but it lacks details. It would be beneficial to explain what is meant by "conformational exchange" and the specific types of dynamics observed. For example, does the trimer undergo transitions between different structural states? Is there any information on the timescales or amplitudes of these conformational changes? Elaborating on the nature of the dynamics will provide a more comprehensive understanding of the protein's behavior.

Additionally, the author might consider including a brief explanation of why understanding the dynamic behavior of the Lcl-CTD trimer is relevant or important in the context of *Legionella pneumophila* colonization and GAG interactions.

Reviewer #4 (Remarks to the Author):

Rehman et al.'s manuscript "The *Legionella* collagen-like protein employs a unique binding mechanism for the recognition of host glycosaminoglycans" describes a study of the *Legionella* collagen-like protein (Lcl) C-terminal domain (CTD). Significantly, they crystallize the Lcl CTD, present a model for intact Lcl, and use molecular dynamics simulations to investigate the binding between the Lcl-CTD and the GAG chondroitin-4-sulphate. The study addresses an important

question related to bacterial adhesion, discovers an unusual arrangement of the Lcl-CTD in a trimer configuration, and focuses on a bacteria that causes illness as the causative agent of Legionnaires' disease. Also, the discoveries made in this paper have implications across other organisms since it is possible that the observed binding mechanism might be more widespread since Lcl homologs are found in other phyla.

In general this manuscript is very well written, with results clearly described and with a thorough description of the methods so that it would be possible to reproduce the experiments described in this paper. The authors should be commended for producing a paper of this quality.

Here I'll provide my technical assessment of the molecular dynamics work in particular. The authors produced docked models using HADDOCK, a standard protein docking tool, between the Lcl-CTD and a variety of oligosaccharides. In particular they found that an Lcl-CTD monomer bound to C4S dp8 oligosaccharide agreed well with their experimental data, and therefore they built a trimer model with one C4S dp8 oligosaccharide was built and simulated using molecular dynamics simulation. They also simulated Lcl-CTD alone to use as a control. Importantly, in their simulations they observed that the trimer complex was relatively stable over the timescales simulated and analyzed. However, although the trimer complex was stable, the oligosaccharide explored different binding configurations on the surface of the trimer, which the authors analyzed using clustering methods, identifying a major and a minor binding mode. The major mode had the oligosaccharide across the middle of the trimer, while the minor mode showed interaction primarily with a single monomer within the trimer complex.

The force field chosen is reasonable for the MD study, and the choices of MD parameters are standard choices for biomolecular simulations. Simulations were of reasonable length, and were appropriately minimized and equilibrated prior to the production phase of simulation. For the Lcl-CTD/C4S systems, 21 150 ns replicas were run, which provides good statistics for the subsequent analysis. The analysis is well described as well, so that the reader can understand how the MD data is interpreted.

I have just a couple of minor comments for the authors about the MD:

The authors mention that the RMSF data for the first 7 amino acids is very large, and indeed this is seen in Panel A of supplementary figure 17. I recommend that the authors also recalculate the RMSF data excluding the first 7 amino acids from the process of alignment, averaging, and RMSF calculation (the N-terminus is expectedly flexible), and then present the data again. In this additional figure panel the authors should show the RMSF on a scale which makes it easier to see the features of the RMSF curve.

The gray vertical lines in supplementary figure 18 are not very visible. Perhaps use a black dashed line to separate the replicas. Also, I would appreciate it if the authors also created a histogram of the data instead of just the time course data which would provide a better sense of the distribution of distance observed between C4S and the Lcl-CTD.

These two changes would make the data more clear for the reader to understand.

Reviewer #1

1) The mode of binding to sulphated GAGs contains a mixture of geometries (eg Fig 7). It would be useful if the authors could comment whether they feel that there is a unique binding mode which they have been unable to determine, or whether it is genuinely fuzzy, for example as described by Forman-Kay's group for Cdc4 binding to multiply phosphorylated Sic1 (<https://doi.org/10.1073/pnas.0809222105>).

We thank the reviewer for this suggestion, and we have now updated the discussion (lines 388-396)

2) Line 67: should substates be substrates?

done

3) Line 209 presumably 0.3 Angstrom?

done

4) Line 314 delete "is"

done

5) Line 423 change were to was

done

6) Line 529 which beamline?

done

7) Line 1017 change peptides to peptide

done

8) Line 1025 change perdeutotared to perdeuterated

done

9) Line 1027 start should read of specific glycine residues

done

10) Line 1028 change glycines to glycine

done

11) Supplementary Figure 7 legend delete the comma after Kratky

done

12) Supplementary Figure 11 legend change is to in

done

Reviewer #2

*1) One of the speculations about the Lcl model is its attachment to the outer-membrane of the host bacterium through an amphipathic helical region (residues 1- 31) illustrated in Fig. 2 panel d. The hydrophobic patch on one quadrant of the helix is flanked by positively and negatively charged residues, which is often seen in helix coiled coils. Have the authors scanned the helix sequence for its propensity to form a coiled coil structure with a complementary helix on the bacterial cell? Has the 31-residue alpha-helical peptide been shown to bind *L. pneumophila*?*

We have carried out bioinformatics analysis on the N-peptide sequence and there is no prediction of a coiled-coil region or indication that it binds a complementary helix on the bacterial cell. However, we have now determined that this region can form trimers (see next comment), so we have included this as potential binding route in the discussion (lines 326-335). We have also confirmed binding to the Lp surface (see next comment).

2) The Discussion section contains a lot of speculation and it would be preferable to make a clearer distinction between what has been established by experimentation and what is speculated to happen. For example, regarding lines 314 and 315: “In this study we have revealed that Lcl is likely binds the bacterial surface via an amphipathic helix motif at its N-terminus, which represents a new mechanism that has not been observed for other T2SS.” Going from ‘likely binds’ to: ‘a new mechanism that has not been observed for other T2SS substrates (Fig. 2d,e)’, is a big stretch. Why not tag the amphipathic helix motif with GFP and see how well it binds to the bacteria.

We thank the reviewer for this suggestion. We have carried out this request and it has added significant further insight into this system. We created a C-term GFP fusion and during SEC observed monomer/trimer states in equilibrium, with trimerisation increasing with concentration. Alphafold produces sensible models for a trimer with the hydrophobic face packing in the core. MD simulations show this is generally stable over 1 μ s, although the helical packing does open. In addition, we developed a new Lp binding assay and have confirmed that Lcl-N-GFP, but not GFP alone, can associate with the bacterial surface. However, it is not clear what form (monomer or trimer) is responsible for surface binding. We have now updated the results (lines 132-149; new Fig 2), discussion (lines 326-335), methods (lines 482-492; 520-522; 544-561) and sup data (Sup Fig 2).

3) The trimeric structure of both the CTD and the collagen-like region (CLR) are consistent. Are there any indications of one trimerization facilitating the other? Which is likely to be the driver of oligomerization?

Although we now see that the N-term as well can form trimers, as with eukaryotic collagens, we expect that the C-term will initiate folding of the collagen region. We have added this in the discussion (lines 357-360).

4) Figure 2. In panel a, the void and total volumes of the size-exclusion column should be indicated in the chromatogram. In panel b, the schematic of the Lcl trimer could show CLR region forming a triple helix rather than parallel strands.

These figures have now been updated.

5) *Given the similarity of the Lcl-CTD monomer to C-type lectin-like domains, could you comment on the number and location of disulfide bonds in the fold? C-type lectin-like domains typically have two or three highly conserved disulfide bonds.*

CTD does not contain any disulfides and this has now been noted in the results (line 186).

6) *Line 314: needs revision*

done

7) *Line 317: Gram-positive, Gram-negative*

done

8) *Paragraph beginning line 413: replace apostrophes with prime symbols.*

done

9) *Some hyphens missing: 1-nm bandwidth; 9-Å distance; carbohydrate-binding mechanism; ligation-independent*

done

Reviewer #3

1) *Page 9, 219: Please comment on the role of R342A in the dissociation of the trimer assembly.*

The author should discuss the specific impact of the R342A mutation on the trimer assembly. Does it lead to the destabilization or disruption of the trimer structure? Any experimental findings or insights on this mutation's effect should be elaborated upon.

We have now provided additional analysis of this mutant and its effect on oligomerisation. This is included in the results (lines 234-236; Sup Fig 12) and discussion (lines 403-416).

2) *Page 10 and Abstract: "Lcl-CTD trimer experiences conformational exchange and is a dynamic structure" - please elaborate on the nature and type of dynamicity. In the text, there's a statement about the dynamic nature of the Lcl-CTD trimer, but it lacks details. It would be beneficial to explain what is meant by "conformational exchange" and the specific types of dynamics observed. For example, does the trimer undergo transitions between different structural states? Is there any information on the timescales or amplitudes of these conformational changes? Elaborating on the nature of the dynamics will provide a more comprehensive understanding of the protein's behavior. Additionally, the author might consider including a brief explanation of why understanding the dynamic behavior of the Lcl-CTD trimer is relevant or important in the context of Legionella pneumophila colonization and GAG interactions.*

We thank the reviewer for this suggestion. We have now moved these statements into the discussion and elaborated on the nature and properties of the dynamicity (lines 403-416; 430-434).

Reviewer #4

1) The authors mention that the RMSF data for the first 7 amino acids is very large, and indeed this is seen in Panel A of supplementary figure 17. I recommend that the authors also recalculate the RMSF data excluding the first 7 amino acids from the process of alignment, averaging, and RMSF calculation (the N-terminus is expectedly flexible), and then present the data again. In this additional figure panel the authors should show the RMSF on a scale which makes it easier to see the features of the RMSF curve. The gray vertical lines in supplementary figure 18 are not very visible. Perhaps use a black dashed line to separate the replicas. Also, I would appreciate it if the authors also created a histogram of the data instead of just the time course data which would provide a better sense of the distribution of distance observed between C4S and the Lcl-CTD.

We thank the reviewer for these suggestions. We have now added a panel to Figure S17 (panel b) to show the RMSF calculated without the first 7 residues. We modified the vertical lines in Figure 18 as suggested and we added a panel to show the cumulative histogram of the distances calculated over all the replicas for each chain.

REVIEWERS' COMMENTS

Reviewer #2 (Remarks to the Author):

The revision of Rehman et al. (NCOMMS-23-43887A) has addressed all the relatively minor concerns that I had with the original manuscript. The authors have done a diligent job in their revisions to answer queries and correct any mistakes. The additional experimentation with the GFP-tagged N-terminal sequence was helpful. This is a significant contribution towards understanding how the bacterial pathogen, *Legionella pneumophila*, can infect its hosts.

Reviewer #3 (Remarks to the Author):

The authors adequately responded to my questions/concerns, and the revised manuscript is now suitable for publication in Nature Communications.

Reviewer #4 (Remarks to the Author):

The revisions of the manuscript in response to my reviewer comments satisfy the questions that I had.